# A structural signature motif enlightens the origin and diversification of nuclear receptors

Brice Beinsteiner [1,2,3,4‡], Gabriel V. Markov [5‡], Stéphane Erb [6], Yassmine Chebaro [1,2,3,4], Alastair G. McEwen [1,2,3,4], Sarah Cianférani[6], Vincent Laudet [7]*, Dino Moras [1,2,3,4]*, Isabelle M. L. Billas [1,2,3,4]*

**1** IGBMC (Institute of Genetics and of Molecular and Cellular Biology), Illkirch, France, **2** Université de Strasbourg, Unistra, Strasbourg, France, **3** Institut National de la Santé et de la Recherche Médicale (INSERM) U1258, Illkirch, France, **4** Centre National de la Recherche Scientifique (CNRS) UMR 7104, Illkirch, France, **5** Sorbonne Université, CNRS, UMR 8227, Integrative Biology of Marine Models, (LBI2M, UMR8227), Station Biologique de Roscoff (SBR), Roscoff, France, **6** Laboratoire de Spectrométrie de Masse BioOrganique, Université de Strasbourg, CNRS, IPHC UMR 7178, Strasbourg, France, **7** Marine Eco-Evo-Devo Unit. Okinawa Institute of Science and Technology, Onna-son, Okinawa, Japan

‡ co-first authors
* vincent.laudet@oist.jp (VL); moras@igbmc.fr (DM); billas@igbmc.fr (IMLB)

**Data Availability Statement:** The data and R script necessary to reproduce Fig 3 are available from Dryad (https://doi.org/10.5061/dryad.kkwh70s48). All other relevant data or accession numbers are

## Abstract

Nuclear receptors are ligand-activated transcription factors that modulate gene regulatory networks from embryonic development to adult physiology and thus represent major targets for clinical interventions in many diseases. Most nuclear receptors function either as homodimers or as heterodimers. The dimerization is crucial for gene regulation by nuclear receptors, by extending the repertoire of binding sites in the promoters or the enhancers of target genes via combinatorial interactions. Here, we focused our attention on an unusual structural variation of the α-helix, called π-turn that is present in helix H7 of the ligand-binding domain of RXR and HNF4. By tracing back the complex evolutionary history of the π-turn, we demonstrate that it was present ancestrally and then independently lost in several nuclear receptor lineages. Importantly, the evolutionary history of the π-turn motif is parallel to the evolutionary diversification of the nuclear receptor dimerization ability from ancestral homodimers to derived heterodimers. We then carried out structural and biophysical analyses, in particular through point mutation studies of key RXR signature residues and showed that this motif plays a critical role in the network of interactions stabilizing homodimers. We further showed that the π-turn was instrumental in allowing a flexible heterodimeric interface of RXR in order to accommodate multiple interfaces with numerous partners and critical for the emergence of high affinity receptors. Altogether, our work allows to identify a functional role for the π-turn in oligomerization of nuclear receptors and reveals how this motif is linked to the emergence of a critical biological function. We conclude that the π-turn can be viewed as a structural exaptation that has contributed to enlarging the functional repertoire of nuclear receptors.

within the manuscript and its Supporting Information files.

**Funding:** This work was supported by the Agence Nationale de la Recherche [Grant Number ANR-2010-BLAN-1234 01] (V.L., I.M.L.B, D.M.), by the Université de Strasbourg (Unistra) (I.M.L.B., D.M., S.E., A.G.M, B.B, Y.C.), CNRS (I.M.L.B., D.M., S.C., A.G.M, B.B, Y.C.) and INSERM (I.M.L.B., D.M., A.G. M, B.B, Y.C.), by the association Alsace contre le Cancer (I.M.L.B.). We thank the Fondation pour la Recherche Médicale [Grant number FRM FDT20170437233] and Eurostars for fellowships awarded to BB. Support and usage of platforms was provided by the French Infrastructure for Integrated Structural Biology [FRISBI, ANR-10-INSB-05-01], Instruct-ERIC, a Landmark ESFRI project (I.M.L.B., D.M., A.G.M, B.B, Y.C.) and by the French Proteomic Infrastructure [ProFI, ANR-10-INBS-08-03] (S.E, S.C.). Financial support was provided by GIS IBiSA and Région Alsace in purchasing a Synapt G2 HDMS instrument (S.E and S.C.). The funders played no role in the study design, data collection and analysis, decision to publish, or preparation of the manuscript.

**Competing interests:** The authors declare that they have no conflict of interest.

## Author summary

The origin of novelties is a central topic in evolutionary biology. A fundamental question is how organisms constrained by natural selection can divert from existing schemes to set up novel structures or pathways. Among the most important strategies are exaptations, which represent pre-adaptation strategies. Many examples exist in biology, at both morphological and molecular levels, such as the one reported here that focuses on an unusual structural feature called the π-turn. It is found in the structure of the most ancestral nuclear receptors RXR and HNF4. The analyses trace back the complex evolutionary history of the π-turn to more than 500 million years ago, before the Cambrian explosion and show that this feature was essential for the heterodimerization capacity of RXR. Nuclear receptor lineages that emerged later in evolution lost the π-turn. We demonstrate here that this loss in nuclear receptors that heterodimerize with RXR was critical for the emergence of high affinity receptors, such as the vitamin D and the thyroid hormone receptors. On the other hand, the conserved π-turn in RXR allowed it to accommodate multiple heterodimer interfaces with numerous partners. This structural exaptation allowed for the remarkable diversification of nuclear receptors.

## Introduction

The nuclear hormone receptor (NR) superfamily includes receptors for hydrophobic ligands such as steroid hormones, retinoic acids, thyroid hormones or fatty acids derivatives [1,2]. This superfamily, which clusters 48 genes in human, is subjected to an intense scrutiny because of the essential role played by NRs in animal development, metabolism and physiology. NRs are important drug targets since dysfunctions of homeostasis and signaling pathways controlled by these receptors are associated with many diseases including cancer, metabolic syndrome or reproductive failure [3].

All nuclear receptor proteins share a characteristic modular structure that consists of conserved DNA and ligand binding domains (DBD and LBD, respectively) separated and flanked by poorly conserved flexible regions [1,2]. Typically, distant NRs exhibit ca. 60% sequence identity in their DBD and 30% in their LBD. The availability of the ligand controls NR activity in space and in time since ligand binding inside a specific pocket within the LBD induces a conformational change of the receptor allowing the release of corepressors, the recruitment of coactivators and the transactivation of target genes [1,2].

Given their importance and also because their long conserved domains are favorable for phylogenetic analysis, the origin of the NR superfamily have been scrutinized for a long time, allowing to define distinct subfamilies [4–6]. Full NRs are specific to animals whereas DBD sequences have been found in the genomes of some choanoflagellates, the closest metazoan relatives [7]. After several lineage-specific events of gene loss or gene duplications, the size of the superfamily ranges from about 20 members in insects to about 48 to 70 in vertebrates, with a specific expansion in some lineages such as nematodes for which more than 260 NR genes are present [8–10].

The analysis of complete genome sequences available in a number of animal species, including early metazoans such as sponges, placozoans or cnidarians have allowed a better understanding of the first step of NR diversification. The observation that sponges, which despite some controversy are believed to be the earliest metazoan phyla [11] contains only two NR genes, called here SpNR1 and SpNR2, have shed a decisive light on the first steps of NR evolution [9]. This has allowed the positioning of the root of the NR tree within subfamily II that in

particular contains the retinoid X receptor (RXR), the hepatocyte nuclear factor 4 (HNF4) and the COUP Transcription Factor 1 (COUP-TF)) and therefore that cannot be considered as monophyletic. This view separates the family into HNF4 on the one hand and all the other NRs on the other hand (Fig 1A). This phylogeny of early NRs now enables the study of the diversification and evolution of the various functions of NRs, such as ligand binding, DNA binding or dimerization. This was done for ligand binding and it allowed to propose that the ancestral NR was a sensor molecule capable of binding fatty acids with low affinity and low selectivity [9,12,13]. However, the same kind of evolutionary analysis has not yet been carried out to study the dimerization properties, a critical aspect of the nuclear receptors functions.

Thanks to their dimerization capability, NRs expanded the range of DNA target sequences through which they regulate target gene expression [1,2,14]. Several distinct dimerization properties have been characterized in NRs among which homodimerization on either palindromic or direct repeat DNA sequences, heterodimerization with RXR as a common partner, and even monomer binding (that is absence of dimerization) through the binding to extended half-site response elements (as depicted in Fig 1B) [14]. The pivotal role of RXR (and of the insect homolog ultraspiracle protein (USP)) in this context has to be pointed out, as it is the promiscuous partner for more than 15 distinct high-affinity liganded NRs, including the retinoic acid receptor (RAR), the thyroid hormone receptor (TR), the vitamin D receptor (VDR), the peroxisome-proliferator-activated receptor (PPAR), the liver X receptor (LXR) or the ecdysone receptor (EcR) in insects. Structural analysis revealed that NRs contain two separable dimerization interfaces, a relatively weak, albeit important interface in the DBD that plays a key role in DNA target site selection [15,16] and another stronger interface in the LBD. The detailed analysis of the LBDs dimerization interface highlighted the rules controlling homo- versus heterodimerization and allowed two functional NR classes to be defined according to their oligomeric behavior [17]. Class I NRs behave either as monomers or homodimers and exhibit a set of conserved residues that form a communication pathway linking helix 1 to the dimerization interface via helix 8. In contrast class II receptors encompass all NRs that heterodimerize with RXR and exhibit a different communication pathway linking the central helices H4/H5 to the dimerization interface via a conserved arginine residue in the loop between helices H8 and H9 [17]. To deepen our understanding of how changes in NR dimerization properties contributed to the diversification of the NR superfamily, we carried out an evolutionary analysis of NR genes, focusing on the evolution of dimerization across the entire NR superfamily. We show here that homodimeric binding was ancestral, whereas heterodimeric and monomeric behaviors evolved later. We further identified a specific structural feature present in helix H7, a so-called π-turn or α/π-bulge, present in RXR and HNF4, as being an ancestral motif critical for the homodimerization of the most ancient NRs. We traced back the complex evolutionary history of this π-turn showing that it was instrumental in the origin of heterodimerization, by allowing a flexible dimerization surface of RXR to accommodate numerous partners with multiple interfaces. The π-turn was originally used for homodimerization, but later was utilized for a different function, namely heterodimerization. This can be considered as a structural exaptation which can be seen as instrumental for the expansion of the repertoire of NR functions.

## Results

### A specific π-turn motif is an ancestral feature of helix H7

Since the first crystal structure of a NR LBD [18,19] more than 800 sets of LBD coordinates have been deposited in the Protein Data Bank. A comparative analysis of these structures with the entire PDB data base showed that the canonical α-helical fold of the LBD is conserved,

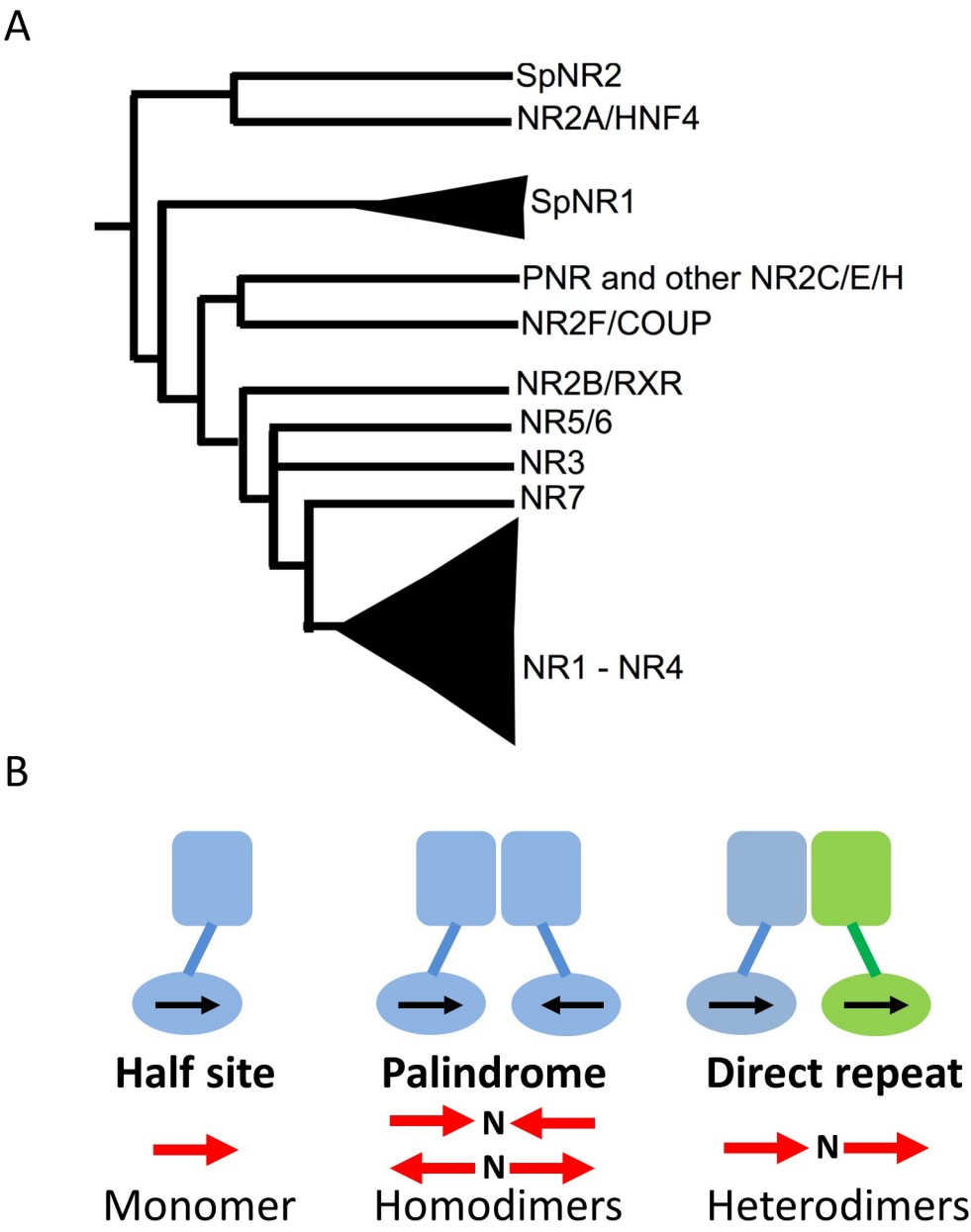

**Fig 1.** (A) Simplified consensus tree of NR phylogeny based on Bridgham et al., 2010 [9]. (B) Some of the diversity found for the different DNA response elements (REs), illustrated here with a monomeric NR on an (extended) half-site; a homodimer on an inverted or an everted RE; and finally a heterodimer on direct repeat RE; N is the number of base pairs in the spacer between the two half-sites.

suggesting strong structure-function constraints during evolution. A peculiar feature emerged from this structural analysis, which the presence of a helical deformation called π-turn or a α/π-bulge within the α-helix 7 of RXR-USP and HNF4 LBDs (RXR, PDB: 1LBD, 6HN6 [18,19]; USP, PDB: 1G2N, [20]; HNF4), PDB: 1LV2, [21] (Fig 2). π-helices and π-turns account for over 15% of all known protein structures deposited in the PDB database [22–25]. The π-type helical structures are thermodynamically less stable than α-helices and are considered to be favored only when they are associated with a functional advantage, typically for interactions with ligands or in the functioning of helical transmembrane domains. The occurrence of π-

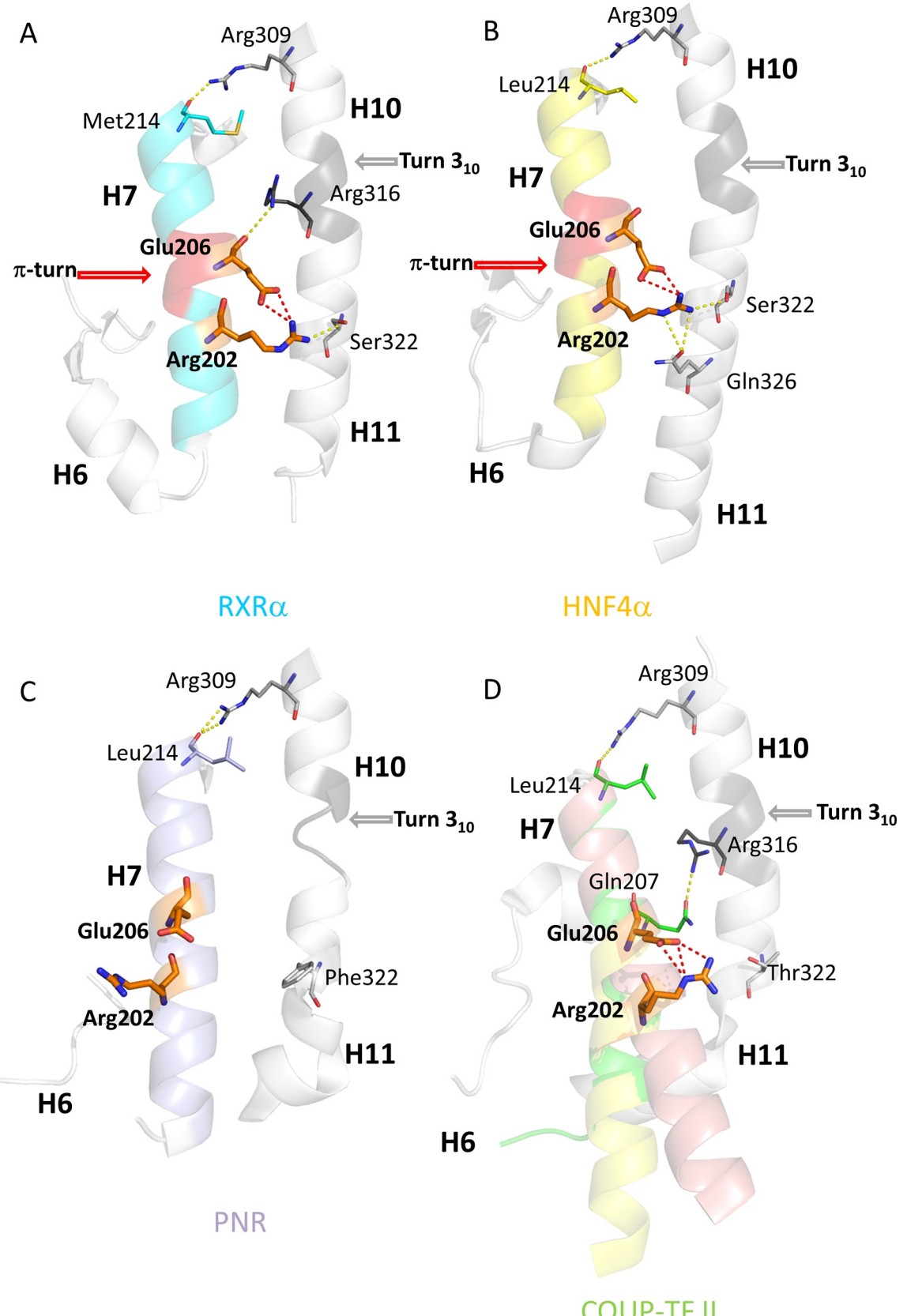

**Fig 2. The environment of the π-turn in helix H7 of nuclear receptors.** The environment of the π-turn is shown for (A) RXRα, (B) HNF4α, (C) PNR and (D) COUP-TFII LBDs. The π-turn is shown in red, the π-turn motif residues (R202 and E206) are shown in orange. The 3₁₀ helix at the H10-H11 junction in dark gray. In all cases, the C-terminal part of helix H7 is anchored to the N-terminal part of H10 by a conserved arginine (R309). Helix H7 is shown in blue and yellow ribbon representation for (A) RXRα and (B) HNF4α. The side chains of the signature motif residues, R202 and E206, and S322 of H11 form a triad of H-bonds (A) and (B). R316 at the H10-H11 junction in **(A)** RXRα and Q326 in H11 of (B) HNF4α complete the set of conserved bonds. For (C) PNR and (D) COUP-TFII, H11 is unstructured and the amino acid shift due to the absence of π-turn only affects the N-terminus of H7, while no major changes occur at the C-terminal part, in line with the structural alignment of the corresponding residues. In (C), PNR is depicted with H7 in light violet. Contacts between H7 and H10-H11 are only observed at the C-terminus of H7. In helix H11, a phenylalanine residue (F322) replaces the serine residue S322 of RXR that is important for interaction with the π-turn. In (D), the original conformation of COUP-TFII is shown in green, the re-refined H7 conformers (see Materials and methods.) are shown in salmon and yellow for the straight and the curved helical conformations, respectively. A threonine residue (T322) in H11 replaces S322 of RXR H11 that interacts with the π-turn. The figures are based on the PDB structures 1DKF for (A) RXRα, 4IQR for (B) HNF4α, 4LOG for (C) PNR and on the PDB structure 3CJW and our re-refined structure for (D) COUP-TFII.

turns in the receptors that are considered to be at the origin of the NR family raises several questions, notably concerning the functional implications of this structural feature.

A conserved RxxxE motif, where the two invariant residues R and E form an intra-helical salt bridge further characterizes this specific conformation. In a π-helical loop, also called a π-turn, the N+4 classical hydrogen bonds of the α-helix are replaced by N+5 hydrogen bonds [25,26]. The π-helical geometry results in the protrusion of the E residue out of the axis of the helix H7 with the two polar residues, E and R, closer to the helices H10-H11. Their side-chains form intricates inter- and intra-molecular interactions, stabilizing the *per se* energetically unfavorable π-helical conformation. The glutamate residue allows the formation of an intra-molecular salt-bridge with the conserved arginine residue of the motif. The arginine residue helps connect helix H7 to helices H10-H11 through binding to a conserved serine residue in helix H11 (S322 on the alignment, S427 in RXRαHS, S1 Fig and Table 1). An additional hydrogen-bond is observed between the π-turn and helices H10-H11. In RXR-USP, the H-bond is formed between E206 (E352 in hRXRα) and R316 in H10 (R421 in hRXRα) (Fig 2A). In HNF4, R202 (R267 in hHNF4α) binds to Q326 of H11 (Q350 in HNF4α) (Figs 2B and S1 and Table 1).

The π-bulge induced shift of residues only affects the N-terminal part of H7. The C-terminal side is anchored by a conserved bond between the carbonyl group of residue M/L214 (H7) and the side chain of R309 (H10). A similar type of interaction pattern prevails for both receptors and leads to strong interactions between H7 and H10-H11. These helices, together with the loop H8-H9 and helix H9, are the main contributors to the canonical NR LBD

**Table 1. Amino acid residue mapping for nuclear receptors considered in this study.** H stands for helices. 'Alignment residue' is the generic numbering used in this study (S1 Fig). The class specific residues are specified by I and II for class I and class II NRs, respectively, with boxes colored in blue and green, respectively, or in cyan for residues common to class I and class II, together with the corresponding residue number [17]. The H7 column with yellow boxes refers to residues of the π-turn, further highlighting the differential conservation of the p-turn.

| Secondary structure | H4-H5 | H5 | H7 | | | | | H8 | loop H8-H9 | | H10 | | | H11 | |
|---|---|---|---|---|---|---|---|---|---|---|---|---|---|---|---|
| Class specific | I (W) | II (E,D) | π-turn motif | | | | | I; II (E) | | II (R) | I (R) | | I; II (R) | | |
| **Alignment residue** | 109 | 111 | 202 | 206 | 207 | 210 | 214 | 220 | 262 | 263 | 309 | 316 | 321 | 322 | 326 |
| **Brelivet numbering** | 40 | 42 | - | - | - | - | - | 50 | 61 | 62 | 93 | 100 | 105 | 106 | - |
| hRXRα | W305 | E307 | R348 | E352 | L353 | K356 | M360 | E366 | D379 | S380 | R414 | R421 | R426 | S427 | K431 |
| hHNF4α | A224 | E226 | R267 | E271 | L272 | P275 | L279 | E285 | D298 | A299 | R333 | L340 | Q345 | S346 | Q350 |
| hCOUP-TFII | W249 | E251 | R293 | E297 | Q298 | K301 | L305 | E311 | D324 | A325 | R359 | R366 | R371 | T372 | S376 |
| hPNR | W257 | E259 | R301 | E305 | T306 | R309 | L313 | E319 | E332 | T333 | R367 | L374 | R379 | F380 | E384 |
| hRARα | C265 | D267 | D307 | A311 | F312 | Q315 | L319 | E325 | D338 | R339 | M373 | K380 | R385 | S386 | K390 |
| hTRα | C309 | E311 | D351 | D355 | L356 | S359 | F363 | E369 | D382 | R383 | F417 | K424 | R429 | M430 | C434 |

dimerization interface. Another interesting observation is worth mentioning: in RXR, the contact between the main chain of E206 (H7, π-turn) and the side chain of R316 (H10) occurs in a place where the α-helical conformation of H10 is locally changed to a short 3(10) helix characterized by N+3 hydrogen bonds. This peculiar 3(10) conformation of H10 is observed for all known NRs structures, except for the pregnane X receptor (PXR) and the steroidogenic factor 1 (SF1) that have classical α-helices (e.g. PXR, PDB: 1ILG, [27]; SF1,PDB: 4QJR, [28]). In order to correlate the presence of the RxxxE motif with the occurrence of a π-turn in H7, we carried out a structure-sequence analysis focused on H7 over several thousands of protein sequences. All available nuclear receptor sequences were taken into consideration. For 49 of them, at least one crystal structure was available. The RxxxE motif in H7 was found to be present in the NR2F group (COUP-TF, seven-up (SVP46/7-UP), V-erbA-related protein 2 (EAR-2)) as well as in the Photoreceptor-specific nuclear receptor (PNR) belonging to the subfamily NR2E (but not in FAX, and the tailless receptors (TLL or TLX)). Whereas no crystal structure is available for SVP and EAR-2 LBD, crystal structures were reported for COUP-TFII (PDB: 3CJW, [29]) and PNR (PDB: 4LOG, [30]). None of these structures exhibits a π-turn conformation or a salt bridge between R and E residues of the RxxxE motif (Fig 2C and 2D).

In PNR LBD, H7 exhibits a canonical α-helical conformation with no visible distortions. No intra-molecular interactions are seen between residues R and E of the motif. The serine residue observed in RXR H11 (S322) that is important for the stability of the π-turn is replaced by F322 in PNR. This residue would generate a steric clash with a π-turn conformer. If a π -helix would be present in PNR, the offset induced by the bulge would change the position of E200 that would then point into the direction of H5-H6, more specifically into a hydrophobic region composed of several leucine residues that would not favor interaction.

The π-turn is also absent in the crystal structure of COUP-TFII [29]. The N-terminal part of H7 is partially disordered and lacks a stabilizing interaction with H11. Furthermore, the neighbouring helix H6 and the upstream connecting β-sheet are not present in the model. A closer inspection of the electron density map suggests that the N-terminal part of H7 could adopt different conformations. Since this part of the protein is critical for our analysis of the RxxxE motif and the structural features associated to it, we refined the protein structure around this location by iterative building of residues in the non-interpreted electron density map followed by crystallographic refinements using PHENIX software (S2 Fig and S1 Table). The newly refined electron density map shows that helix H7 is more extended at its N-terminal side and adopts two conformations, a regular straight α-helix and a curved one bent at the level of the π-turn. The C-terminal parts of the two helical conformations overlap nicely, while their N-terminal ends are 6 Å apart. These two conformations are in equilibrium in the crystal, alternating between nearest neighbour molecules to ensure optimal packing and are likely to be natural conformations. The dynamics of H7 resulting from the lack of stabilization through interactions with H11 promotes the adaptability to packing constraints with a subsequent disorder of this subdomain. The intra-helical salt bridge between the side chains of the conserved arginine R202 and glutamic acid E206 of the motif is conserved, but rotated to a position where no interaction between the motif and H10-H11 can take place, since the shift induced by the absence of the π-turn prevents E206 from binding R316. Instead the connection is made with its neighboring residue Q207 (Q298 in hCOUP-TFII). A threonine residue that does not interact with H7 residues replaces the conserved serine residue in H11 that stabilizes the π-turn in RXR-USP and HNF4. In addition, no interactions are seen between H7 and H5-H6. Altogether, our analysis of the re-refined crystallographic structure of COUP-TFII unambiguously demonstrates that the RxxxE motif present in the sequence of this receptor is structurally associated neither with a π-turn in helix H7, nor with a 3(10) helical turn as suggested in the original structure [29].

## The π-turn motif is ancestral and has been lost several times independently

The analysis of sponge nuclear receptor sequences show the presence of the RxxxE motif in helix H7 of SpNR1, but not of SpNR2. SpNR1 is associated with the group of nuclear receptors NR2B, C, D, E, F as well as NR3/4/5/6 subfamilies, while SpNR2 belongs to the HNF4-like subfamily. The markers of dimerization for class I and class II corroborate this interpretation (E5, W40, K/R55, R/K93, R105 for class I NRs; E/D42, R62, H/R/K90 for class II NRs and E50 and R105 universally conserved) [17]. Indeed, SpNR1 encompasses all of the class I markers, while in SpNR2, two class I markers (W40 and R105) are missing. Interestingly, the same class markers are absent in HNF4. Homology modelling of SpNR1 using a reference panel of nuclear receptor structures, which in majority do not have a π-turn, indicates the presence of a π-turn in 98% of the generated models (see Materials and methods). Furthermore, when SpNR1 replaces RXR in the structures of homodimers or heterodimers, the essential dimeric interactions are conserved. This suggests that the essential distinguishing features of RXR that can exist as a homodimer as well as a heterodimerization partner were already present in SpNR1.

In order to understand the evolutionary dynamics of the π-turn motif conservation, we plotted the presence of the π-turn motif, as well as that of the RxxxE motif on a phylogenetic tree of NR sequences (Figs 3A and S3). Our tree topology is fully consistent with previous studies [8,9]. The tree allows to robustly position most NR subfamilies, even though a major unresolved trichotomy still subsists concerning the branching of the NR3 and NR5/6 families relative to the robust NR7/NR4/NR1 cluster. Interestingly, our current sampling regarding sponges and other early metazoans sequences indicates that, within SpNR1, a lineage-specific amplification has occurred in calcareous sponges, leading to four distinct paralogues (numbered P1 to P4 in S3 Fig). Our analysis therefore includes the whole currently known diversity of early NRs (see S3 Fig).

Taken together, these data suggest that the π-turn, and its associated RxxxE motif were present ancestrally in the primordial nuclear receptors and lost in several rapidly evolving lineages of basal NRs (*e.g.* sponge SpNR2, some paralogous sponge SpNR1), as well as in the major derived NR subfamilies (*i.e.* NR2EF, NR3, NR1 etc.).

## The ancestral NR activated transcription as a homodimer

Nuclear receptors exhibit three different modes of oligomerization: homodimer binding (*e. g.* steroid receptors or HNF4), heterodimers with the promiscuous partner RXR (e.g. TR, RAR, LXR or PPAR) and monomer binding (e.g. SF1 or Rev-erb) [1,31] (Fig 1B). It is important to note that these modes of binding are not mutually exclusive. For example, homodimer formation has been demonstrated for Rev-erb which can also bind to DNA as a monomer [32]. Similarly, RXR can form either homodimers or heterodimers. This oligomeric behavior is related to the mode of binding to DNA, since response elements are derivatives of a canonical sequence (A/GGGTCA) that can be modified, extended or duplicated therefore offering a large palette of possible NR-selective binding modes [31]. As mentioned above, the final oligomeric status is thus the result of the interplay between the strong dimerization interface in the LBD and a weaker one in the DBD which is crucial for response element selection [1]. To trace back the evolutionary history of the dimerization abilities of NRs, it is therefore necessary to fully disentangle the DNA binding and response element selection from the oligomeric status. For this reason, we focused our analysis on the major dimerization interface of the isolated LBD.

To understand the evolution of dimerization of nuclear receptors, we mapped the dimerization patterns of each receptor, using the four states already defined [33–36] on a simplified version of our updated phylogeny, which is fully consistent in its topology with previous

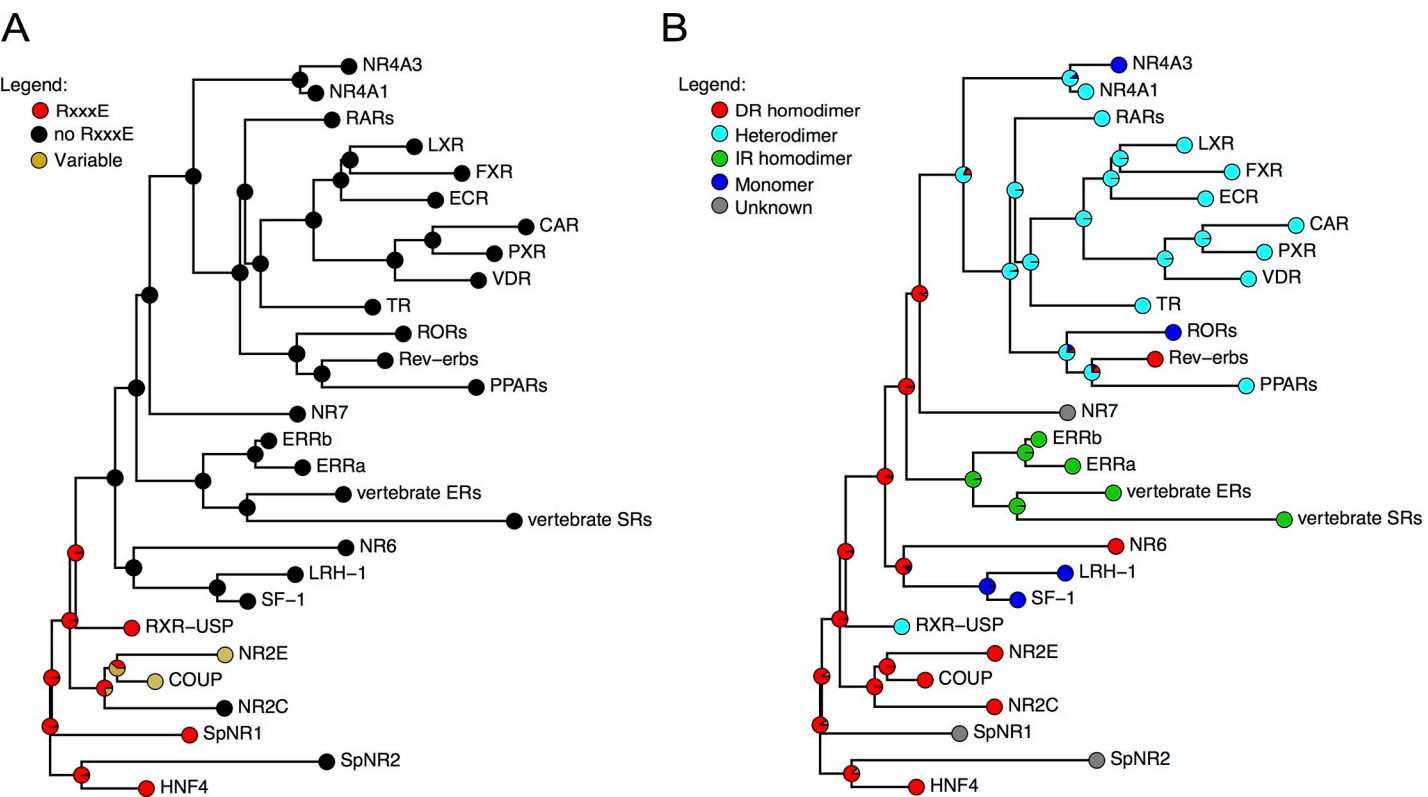

**Fig 3. Evolution of the RxxxE motives compared to the dimerization modes during the history of the NR family. A.** Evolution of RxxxE motives, as reconstructed using ancestral mapping. **B.** Evolution of dimerization modes. Source files and script are provided in dryad under the following doi:10.5061/dryad.kkwh70s48.

publications based on a similar dataset [8,9]. We adopted a conservative strategy in that when no experimental data was available, we coded the relevant oligomerization ability as unknown even if clear class I or class II residues can safely indicate the dimerization mode [17].

The ancestral state reconstruction for every node of the phylogeny illustrates successive complexification of the binding mode. Ancestrally, the binding mode is that of a homodimer, then only a heterodimer binding mode emerged once at the basis of the NR1 and NR4 families, while the monomer binding mode appeared several times independently from either from DR-homodimer or from RXR-heterodimers (Fig 3B).

## The π-turn residues are required for HNF4 biological function

According to the NR partition into class I (monomers and homodimers) and class II (heterodimers) NRs, RXR-USP and HNF4 belong to the class I. Class I differentially conserved residues (i.e. residues strictly conserved in class I and strictly absent in class II NRs) define a class-specific interaction pattern that connects together H1 to H8 and H8 to H10, thereby networking the ligand binding pocket to the dimerization interface [17]. Examination of the class I-conserved residues in HNF4 indicates that this receptor is an outlier of the class I NRs. In fact, two class I invariant residues, W109 (W40 in the alignment given by [17]) and R321 (R105) are not conserved for HNF4, W109 being replaced by an alanine residue and R321 by a glutamine residue (Table 1). In HNF4, the residues A109 and Q321 are essentially conserved from cnidarians to mammals. In contrast, in sponge spNR2, W109 is replaced by a valine residue and R321 is mainly replaced by lysine or tyrosine residues but in a context lacking the π-turn. Note that these two residues have important structural and functional roles. W109 is located at the

junction of H4-H5, a highly conserved structural feature of the class I NR family and an interaction hot spot for ligands. It was shown to be involved in a ligand-dependent allosteric mechanism in RXR [37]. The arginine residue R321 in H10 is an important residue of the dimerization interface highly conserved for all NRs, except in HNF4 and the oxosteroids subgroup (the androgen (AR), glucocorticoid (GR), mineralocorticoid (MR) and progesterone (PR) receptors). In the latter family, and only there, this mutation is associated with the mutation of the residue E111 (E42) that is normally strictly conserved in the whole NR family. In RXR, but not in HNF4, this amino acid residue binds to R321 and contributes to the stability of the homodimer. Altogether, the mutation of the two highly conserved residues W109 and R321 in HNF4 highlight the early divergence of this receptor from the rest of the family.

The analysis of the crystal structures of the HNF4α homodimer shows that a large contribution to the stability of the homodimer comes from the unusual stacking of the tryptophan residue W325 (W349 in hHNF4α) in H10 with the corresponding residue of the other subunit (Fig 4A). These residues and the corresponding contacts they form are specific to this receptor family. Furthermore, several other residues of H10 belonging to one subunit contact helix H9 and the loop H8-H9 of the other subunit, thus forming a strong interaction network. In addition, the π-turn residue E206 in one subunit interacts with the region H10-H11, thereby forming intermolecular stacking interactions with D262 (D298 in hHNF4α) in the loop H8-H9 of the other subunit (Fig 4A). In summary, we observe an intricate and unusual interaction network that involve residues of the π-turn as well as helices H9 and H10 in both subunits. When we compared HNF4 to RXR, the numerous contacts (H-bonds, VdW. . .) that link together the two LBD subunits result in a larger buried surface at the dimer interface, consistent with an energetically more stable oligomer.

The functional importance of the π-turn of HNF4 to the homodimerization process was demonstrated in earlier studies, where the π-turn residues R202 and E206 were mutated and the functional consequence assessed [38]. In this work, it was shown that removing the charges of R202 and E206 impairs dimerization of the protein in solution and affect the HNF4α transcriptional activity in a variety of different cell lines. The impairment on transcriptional activity is even larger for the deletion mutant ΔE206 (E262 in hHNF4α), which was also shown to be less efficient in recruiting transcriptional partners, such as SRC-1 and PGC-1. To correlate with biological effects, we searched the library of human HNF4 mutations reported for MODY1 syndrome and for various cancers that feature HNF4 somatic mutations. We found a small number of somatic mutations in the π-turn motif, especially affecting R202 (R267 in hHNF4α), suggesting that this residue is indeed important for the biology of HNF4 in humans (HGMD database [39]).

Two specific features could explain absence of HNF4 heterodimers. First, the numerous H-bonds linking H9 and H10 of the LBD partners, mostly absent in RXR, are largely responsible for the strength of HNF4 homodimers, and much more stable than RXR ones (Fig 4A and 4B). Interestingly in RXR heterodimers, the number of bonds between helix H10 of RXR and helix H9 of the partner NR increases significantly (Fig 4C). Second, the mutation of a class I and II marker, R381 in RXR, Q321 in HNF4, is another remarkable feature. In RXR homo and heterodimers, R321 is bound to serine 322, conserved in most class II partners.

## The π-turn motif is critical for RXR homodimer formation

In contrast to the strong and intricate homodimerization interaction interface of HNF4, the dimerization interface of RXR dimer involves less contacts, as shown in Fig 4B. The scarcity of the interactions between the two subunits of the homodimer suggests a less stable dimer. Few interactions are observed between H10 of one subunit and H9-H10 and the loop H8-H9 of the

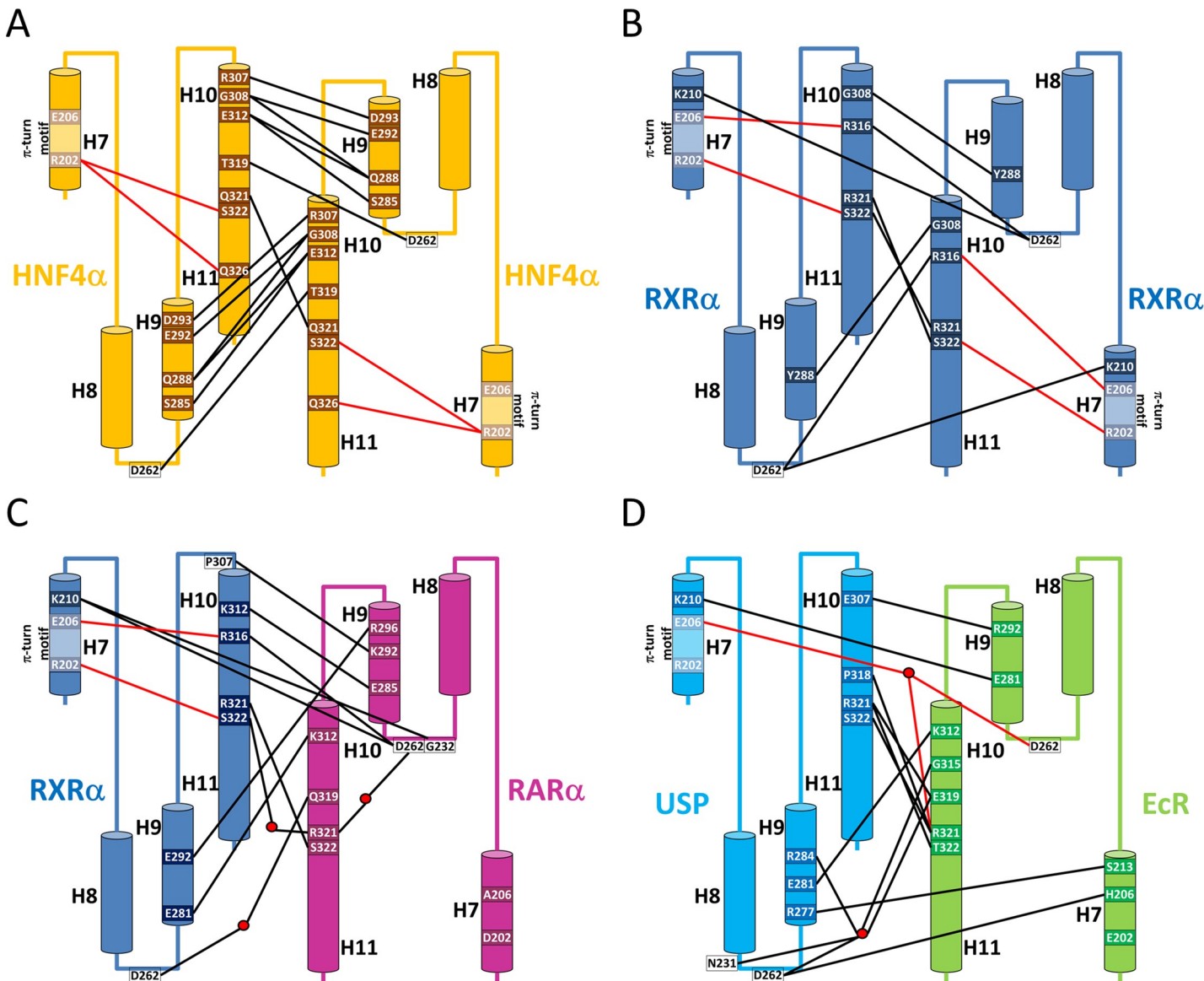

**Fig 4. Schematic representation of the dimer stabilizing bonding interactions.** The set of α-helices H7 to H11 of each subunit of the LBD dimer is represented by cylinders. Water molecules are shown by red dots. Dimerization bonds between the two LBD subunits are represented by black lines and bonds between residues R202 and E206 of the π-turn motif and residues at the dimer interface are shown by red lines. Shown in the scheme are bonds which are most frequently observed in the majority of the available structures. (A) HNF4α LBD. The dimer is symmetric and mainly stabilized by cross-contacts between helices H9 and H10. The π-turn contributes in an indirect manner to the dimerization interface through internal bonds with residues of helix H11. (B) RXR LBD. The dimerization is symmetrical, but less stabilizing bonds can be observed as compared to the HNF4 case. Interactions are also more uniformly distributed over the entire interface, but helix H9 is weakly involved in the dimerization interface, whereas the π-turn is directly and strongly implicated. (C) RXR-RAR. The dimer is asymmetrical, and helices H9 are more strongly involved in the dimerization interface. The helix H7 of RXR directly contributes to the dimer interface through its π-turn. There is no contribution of RAR helix H7. (D). USP-EcR. The interface is asymmetrical, but less asymmetric than for the different heterodimers formed with RXR. Note the unusual role played by helix H7 of EcR in the dimerization process [48].

other subunit. Importantly, the π-turn residues play a crucial role in the stability of the homo-dimer. R202 and E206 are both involved in the dimerization interface. R202 link together S322 (H11) in the same subunit to R321 in H10 of the other subunit. E206 links together R316 (H10) found in the same subunit to D262 located in the loop L8-9 of the other subunit. The latter residue further interacts with K210 located at the C-ter of the π-turn. Altogether, the

structural analysis shows that the π-turn residues are strongly involved in the homodimerization interface.

To assess the functional importance of the π-turn residues for RXR homodimerization, we sought the effects of mutating the critical residues of the π-turn motif on the dimerization behavior of RXR. To address this question, we chose to mutate E206 (E352 in hRXRα) of the RXXXE motif of RXRα LBD either into an alanine residue or to delete it completely from the LBD protein construct and relied on biophysical methods, including analytical size-exclusion chromatography (SEC), analytical ultracentrifugation (AUC) and native electrospray ionization mass spectrometry (ESI-MS) for the analysis of the oligomeric status of the wild-type and mutant proteins. In addition, we carried out molecular dynamics simulations of wild-type and mutant receptors to gain insights into the stability of the dimeric species.

The analytical SEC analysis was carried out using a S200 10/300 Superdex column by injecting the different proteins after the affinity purification step. The corresponding chromatograms, shown in S4A Fig, reveal notable differences between wild-type (wt) RXRα LBD and E352A and ΔE352 mutant constructs. The three proteins have a peak in common at an elution volume that roughly corresponds to the exclusion volume of the column (called void in S4A Fig), and therefore to large oligomeric protein species. Two additional peaks are observed for wtRXRα LBD (called peak1 wt and peak2 wt in S4A Fig and indicated with red and blue symbols, respectively), whereas only one additional is seen for the mutants RXRα LBD peak (called peak mut in S4A Fig, and indicated with cyan and grey symbols for E352A RXRα and ΔE352 RXRα, respectively), with a similar elution volume. This indicates that the two mutant LBD constructs behave differently compared to wtRXR LBD and lack the larger species that compose peak 1 of wtRXR LBD. Since all SEC peaks correspond to pure protein samples, as shown in the SDS-PAGE gel in the insert of S4A Fig, the difference in the size of the protein species composing each peak can solely be attributed to different protein oligomerization states and not to any co-purified contaminant species.

To further identify the SEC-separated species, SEC was online coupled to native ESI-MS for accurate oligomeric state assessment [40]. SEC-native MS analysis of wtRXRα LBD reveals two peaks, as shown in Fig 5. The first peak (shown in red in the inset of Fig 5) consists of tetramers, whereas the second peak (shown in blue in the inset of Fig 5) is composed of dimers and monomers (Fig 5 and S2 Table). In contrast, the main peak of both RXR mutants corresponds to dimeric and monomeric species only, while no tetramers are detected (Fig 5 and S2 Table). Of note, under strictly identical experimental and instrumental conditions, the ΔE352 RXRα mutant exhibits more dimers than the E352A RXRα mutant, which might suggest a slightly increased homodimer stability for ΔE352. Altogether, the MS analysis indicates that mutating E352 of the RxxxE motif of RXRα LBD dramatically impairs noncovalent tetramer formation when compared to the wtRXRα LBD. However, we still observe a low abundance population of mutant RXRα LBD dimers in a large crowd of monomers. As strictly identical SEC columns could not be used off line and in-line with native MS, we further collected SEC peaks obtained on a S200 10/300 Superdex and analyzed the fractions by native ESI-MS (S5A–S5D Fig and S3 Table). Again, native MS data analysis indicates that the wtRXRα LBD sample is composed of noncovalent tetramers and monomers (only low intensity dimers are detected) (S5A and S5B Fig and S3 Table), whereas mutant RXRα LBD samples do not exhibit any tetramer species, but rather a mixture of monomeric and dimeric populations (see S5C and S5D Fig and S3 Table).

We further analyzed the RXR SEC fractions by analytical ultracentrifugation (AUC). The AUC data are summarized in S4B Fig that shows the differential sedimentation coefficient distribution c(S) as a function of the sedimentation coefficient S. Two c(S) peaks are observed for the sample corresponding to SEC peak 1 of wtRXRα LBD shown in red in S4A Fig and only

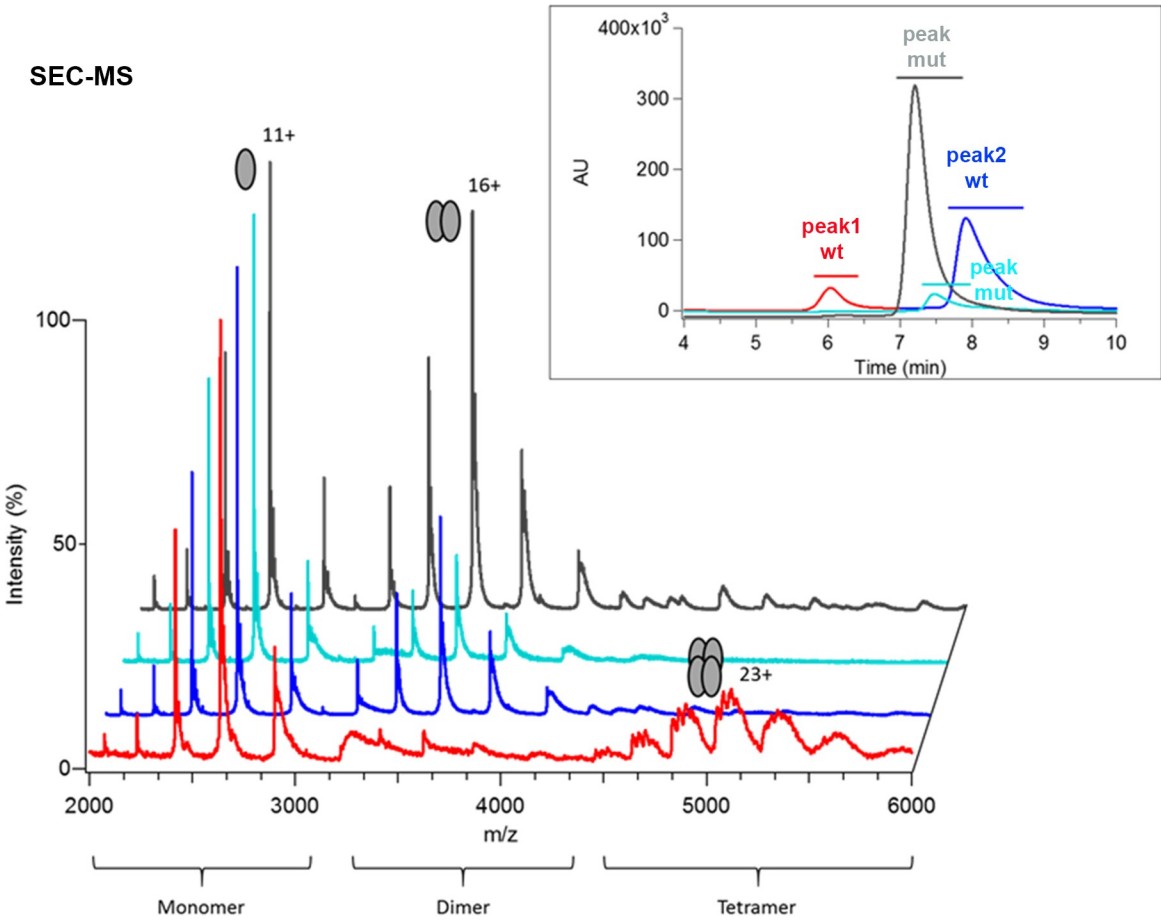

**Fig 5. Oligomeric status of wild-type (wt) RXRα LBD and mutants, where the conserved Glu residue of the π-turn motif is mutated to Ala (E352A) or deleted (ΔE352).** Size-exclusion chromatography (SEC)-coupled native mass spectrometric analysis for wt RXRα LBD (red for the first peak and blue for the second peak), E352A RXRα LBD (cyan) and ΔE352 RXRα LBD (grey). The insert depicts the size-exclusion chromatograms of wt RXRα LBD and of mutants RXRα LBD (with the same color code). The region of the SEC peak considered for the integration of the mass spectra is shown as a line above the SEC peak. For wt RXRα LBD the tetrameric species is seen at the beginning (I) of the SEC chromatogram. In contrast, monomeric species is observed for all SEC peaks together with dimeric species. For mutant RXRα LBD species, no tetramer is observed in the SEC chromatogram and in the corresponding mass spectra where mainly monomeric and dimeric species are observed.

one peak for the other samples (shown in blue, cyan and grey in S4A Fig). Detailed examination of the sedimentation data shows that for wtRXRα LBD, the SEC peak 1 is a heterogeneous sample with several species in dynamic equilibrium, including a majority of tetramers and smaller species down to the monomer, whereas the SEC peak 2 is composed of a mix of monomers and dimers. In the case of the RXR mutants, the AUC data analysis indicates that the samples consist mostly of monomers and a slight amount of dimers. Thus, the two peaks observed in the differential sedimentation coefficient distribution c(S) correspond to the tetrameric species for large S value and essentially to monomer species for the SEC peak 2 of wtRXRα LBD. For the mutants, monomeric species prevail, but the larger width of the c(S) peak suggests the formation of rapidly associating/dissociating dimers from the larger monomer pool. Importantly, no tetramer is observed for the mutants RXRα LBD E352A and ΔE352. The AUC results show that monomers and tetramers are the main species of wtRXRα LBD, in full consistency with SEC-native MS observations (Figs 5 and S5A–S5D) native gel electrophoresis (S5E Fig). Note that a unique band is observed in the native gel of the mutant RXRα LBD

species which might be attributed to the rapidly exchanging monomer/dimer species or to the dominant dimeric fraction, as observed in the AUC and MS analyses.

Altogether, the biophysical data indicates that noncovalent tetramer formation is impaired for the RXRα LBD E352 mutants, in stark contrast with the wild-type protein. The RXRα tetramer is composed of a non-covalently bound dimer of RXRα homodimer and importantly represents the main reservoir of RXRα homodimer available in the cell, as shown *in vivo* and *in vitro* [41,42]. It was shown that disruption of the tetramerization interface of RXRα by mutating conserved phenylalanine residues in helix H11 (depicted in S5F Fig) results in transcriptionally defective protein, without affecting the overall fold of the protein, nor ligand binding, dimer formation or DNA binding. Here, strikingly, we show that the mutation or the deletion of E352 in H7 impairs tetramer formation, by destabilizing the mutant RXRα homodimer species. This residue is far from the tetramerization interface composed of helices H3, H11 and H12 (S4F Fig) [43,44]. However, residues of the π-turn interact with helix H11 and help stabilize its conformation. The mutation of the conserved Glu residue of the π-turn does not prevent homodimer formation since the interface also encompasses other mostly conserved and hydrophobic residues [45]. However, it is likely to destabilize helix H11 and as a consequence to weaken the homodimer interface, enough to lead to the destabilization of the tetramerization interface, as observed experimentally.

We finally carried out Molecular Dynamics (MD) simulations to investigate whether the propensity for dimerization within RXR is affected. MD simulations of 50 ns were thus performed starting from a 1.9 Å resolution crystal structure of HsRXR LBD (PDB: 1MVC [46]. Three sequences were used, including the E357A and ΔE357 mutants. The total binding free energies were then calculated for each complex. The results, shown in S4 Table, suggest that wt RXRα LBD is the most stable homodimeric complex, followed by ΔE352 RXRα LBD and E352A RXRα LBD. Examination of the resulting structures after the MD simulations shows that for the E352 mutants, the contacts between H7 and H10 of one subunit and the loop H8-H9 and H9 of the other subunit are dramatically weakened (S6A–S6D Fig). Altogether, the biophysical characterization and MD simulations suggest that the E352 deletion and mutation has a destabilizing action of RXR homodimeric association, hampering tetramer formation for both mutants.

## The π-turn allowed RXR to evolve as a promiscuous partner for heterodimerization

In contrast to HNF4α, RXRα can form heterodimers with NR partners. In all the cases, the heterodimerization interface is always asymmetric, whereby helix H7 of RXR is closer to the loop H8-H9 of the partner than the reverse. An intricate network of interactions spans the entire interface between RXR and its NR partner that involve helices H7, H8 and H9 and the loop H8-H9. The observation is consistent with experimental data indicating that the heterodimers are more stable than the RXRα homodimer [47]. The asymmetry of the dimerization interface has a direct impact on the number and the type of interactions. By taking RXRα/RARα as an example (PDB: 1DKF [48]), we observe a scarce number of interactions between the loop H8-H9 of RXRα and helix H7 of RARα, while numerous interactions are seen in the reverse situation, *i.e.* between the loop H8-H9 of RARα and H7 of RXRα, and in particular its π-turn residues R202, T205, E206 and K210 (Fig 4C). The direct or water-mediated interactions between the latter residues and residues of the loop H8-H9 of RARα, D262, Q264 and D265, are made possible by the mere presence of the π-turn in RXRα, whereas the absence of π-turn in the partner LBD prevents the establishment of most bonds. Thus, the generation of the strong asymmetry in the dimerization interface is highly dependent on the presence of the

π-turn in RXR and its concomitant absence in partner NRs. The asymmetry of the heterodimer together with the involvement of class conserved residues of the RXR partner in the heterodimer interactions likely favored RXR as a common dimerization partner and led to the emergence of class II NRs as partners of RXR.

Finally, from the evolutionary point of view, it is interesting to consider the well-known ecdysone receptor, which is found in insects and other arthropods, in particular in insects and which is made of a heterodimer between EcR and USP, the ortholog of RXR. Several crystal structures of EcR/USP-RXR LBDs are available from different insect species [20,49–51]. All of the structures exhibit an asymmetric heterodimeric interface, just like vertebrate NRs, with a similar interaction pattern as seen in the previous example of RARα/RXRα. In particular, helix H7 of USP-RXR that encompasses the π-turn makes direct and water-mediated interaction with the loop H8-H9 of EcR (Figs 4D and S7A). On the other hand, analyses of the structures indicate that, depending on the insect species considered, contacts between the helix H7 of EcR and USP-RXR may vary enormously [20,49–51]. In the more basal insect species, such as the beetle *Tribolium castaneum* (Tc) (Coleoptera) and the silverleaf whitefly *Bemisia tabaci* (Bt) (Hemiptera), no or few contacts are observed (none for Bt and one bond between H441 in H7 from EcR to Asp 325 in the loop H8-H9 of USP). In more recent species, such as the moth *Heliothis virescens* (Hv) (Lepidoptera), in stark contrast, numerous bonds are observed linking the helix H7 of EcR and the loop H8-H9 and the helix H9 of USP-RXR (S7A Fig). The origin of this difference between species comes from the position of the loop USP-RXR H8-H9 which is close enough for interaction with EcR in Hv, but not in Tc and Bt (S7B Fig). This discrepancy reflects more profound differences in the overall structure of USP-RXR among the different species [49,52]. In fact, USP-RXR of the basal insect species are more similar to the mammalian RXR than to sequences of USP-RXR of more recent species that encompass the Lepidoptera (moths and butterflies) and Diptera (flies, mosquitos) groups. Therefore, the peculiarity observed for HvEcR/HvUSP-RXR merely reflects the high evolutionary divergence of Lepidoptera and Diptera compared to the other clades [52–54]. However, the analysis of the more recent species EcR/USP-RXR LBDs suggests that independently of the existing interactions made between H7 of EcR and USP-RXR, the asymmetry of the dimerization interface that is dependent of the presence of the π-turn in USP-RXR remains a conserved feature. Altogether, the analyses of the EcR/USP-RXR structures nicely illustrate the evolutionary conservation of the heterodimerization interface, its asymmetry and the involvement of the π-turn into the dimerization mechanism.

To substantiate our hypothesis for the role played by the π-turn in the heterodimerization, we experimentally characterized the heterodimers between RXRα LBD and PPARα LBD for the wild-type and the E352 RXRα mutants. We used SEC-coupled to native MS to relatively quantify the heterodimeric PPARα/RXRα population in the complex mixture between RXRα (wt or mutant) and its partner PPARα LBD. Fig 6 summarizes the relative abundances of monomeric and heterodimeric species as deduced from native MS results (S8 Fig). All RXRα construct (wt but more interestingly also E352 mutants) allow formation of PPARα/RXRα heterodimers (Figs 6 and S8, together with the presence of monomeric RXRα and PPARα. However, there is a strong reduction in the relative PPARα/RXRα heterodimer population between PPARα/wtRXRα and PPARα/E352 mutants (Figs 6 and S8), along with increased amounts of free monomeric RXRα detected.

## Discussion

### The π-turn clusters a crucial interaction network in basal NRs

In this paper, we focused on the π-turn motif which is present in nuclear receptors located at the base of the NR evolutionary tree. The structural importance of this motif that clusters a

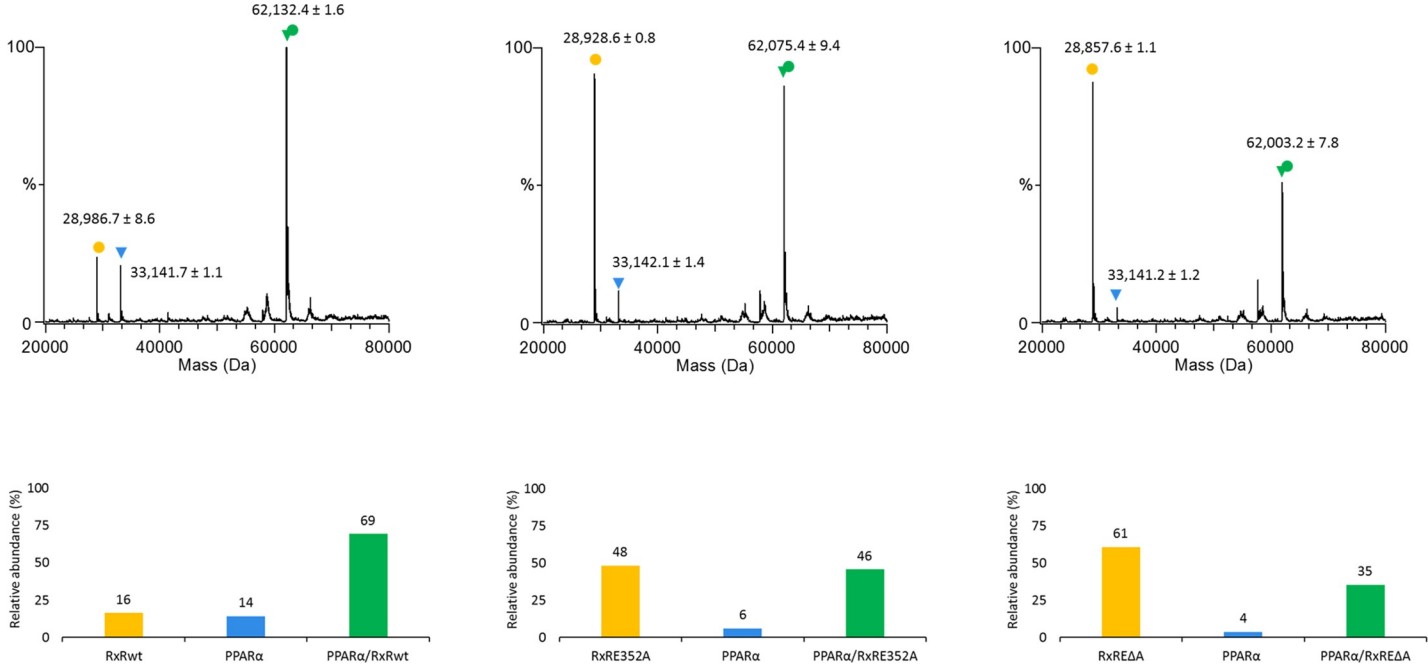

**Fig 6. Heterodimerization capacity of wild-type (wt) RXRα LBD and mutant E352A and ΔE352 RXR LBD with PPARalpha.** Size-exclusion chromatography (SEC)-coupled native mass spectrometry (MS) analysis for complex mixture of PPARα LBD with either (A) wt RXRα LBD, (B) E352A RXRα LBD or (C) ΔE352 RXRα LBD. The isolated RXRα LBD (wt or mutants) is depicted in yellow, isolated PPARα LBD in blue and heterodimeric PPARα/RXRα LBDs in green. Mass spectra obtained by deconvolution of the raw data (shown in S6 Fig) for the three different complex mixtures are shown, together with the quantification of the species from the SEC-native MS analysis of the PPARα/RXRα LBD complex mixture, shown below in the form of histograms.

network of amino acid residue interactions, its complex evolutionary history and conservation in basal NRs strongly suggest that it is a key structural element for the functional diversification of NRs. Our structural analysis reveals that the π-turn, located within helix H7, is always associated with the presence of an RxxxE motif. Furthermore, our 3D homology modeling study allowed us to infer the presence of a π-turn in NRs that harbor the RxxxE sequence motif, but for which no structural and functional data are available. As a result, we observed the presence of a π-turn in HNF4 of bilaterians and basal metazoans, such as cnidarians and placozoans (*Trichoplax*), as well as in RXRs of bilaterians and cnidarians and in SpNR1. The latter case is particularly interesting, since SpNR1 together with SpNR2 represent the only NRs found in sponges. These two receptors, which are used to root the NR superfamily tree, are considered to be the most basal NRs and thus define the two major subdivisions in the NR evolutionary tree, one containing SpNR2 and HNF4 (NR2A) and the other one containing SpNR1 and all the other NRs (Fig 3A) [9].

Our phylogenetic analysis enabled us to hypothesize that the π-turn is an ancestral motif that was present early in the primordial NRs and that was further differentially lost at least five times during the NR evolution. Due to the similarities between the π-turn present in the structures of RXR and those of HNF4, we strongly support the "π-turn early" scenario, rather than the alternative "π-turn late" scenario of the late independent origin of the π-turn in HNF4s, RXRs and SpNR1 (Fig 3A). Based on our analysis, we inferred that a π-turn similar to those seen in HNF4 and RXR should be present in SpNR1. A crystallographic study of SpNR1 LBD would be the ideal test for our hypothesis.

The structure-sequence analysis of NRs that exhibit a RxxxE motif indicates that among all NRs whose structure is known, only RXR-USP and HNF4 possess a peculiar π-helical

geometry. This intrinsically unstable π-helical conformation requires strong stabilizing interactions between the N- and the C-terminal parts of H7 and neighboring regions in its molecular environment to hold together this topological feature. Importantly, the π-turn of H7 thus gives rise to specific intricate interactions with helices H10 and H11, both being crucial element of the dimerization interface. As a matter of fact, the junction between the two helices H10-H11 encompasses a 3(10) conformation and a specific leucine rich sequence (LLLXXL or LLXXL) at the N-terminal part of H10. These structural features, which induce a kink in the region of helices H10-H11, make possible crucial and complementary interactions between the π-turn conformer of H7 and H11, notably between the arginine residue of the RxxxE motif and the serine residue of H11. Therefore, the π-turn is at the heart of the network of interactions present in RXR and HNF4 from the origin for the stabilization of the LBD and for the formation of a stable homodimerization interface ([Fig 4]). These interactions allowed the ancestral receptor to bind DNA response elements as a dimer in a cooperative manner, an ability that increased the DNA binding site selectivity. It is important however to emphasize that the π-turn is not necessary for the homodimerization of all nuclear receptors, but only for RXR and HNF4. In fact, steroid NRs, such as the estrogen receptor (ER) and the estrogen-related receptor (ERR) homodimerize in the absence of π-turn and RxxxE motif, making use of the same secondary structural elements for building the dimerization interface as RXR and HNF4. From an evolutionary point of view, ER and ERR evolved in a way such as they underwent compensatory mutations leading to the disappearance of the RxxxE motif, but conserving most of the other interacting residues. On the other hand, the later evolved oxosteroid nuclear receptors (AR, GR, MR and PR) are different in their dimerization properties. Their ligand binding domain does not dimerize in the same manner as ER and ERR. In fact, there is a marked sequence difference of the residues at the interface compared to the whole nuclear receptor family and the presence of an additional conserved region at the C-terminal end of the LBD that hampers the oxosteroid receptors to dimerize in a classical way [1,55] that still needs to be uncovered.

In protein structures, π-helices and π-bulges are often associated with a specific function, making them powerful markers of protein evolution [23]. An accepted hypothesis about the emergence of π-bulges is their frequent implication as ligand binding site contributors such as in GPCRs (van der Kant and Vrient, 2014). In the case of NRs, a direct association with ligand binding is rather unlikely. For example, both apo and holo crystal structures are available for RXR and, importantly, show no significant differences in the π-turn environment. Note that the *in vivo* relevance of RXR ligands, such as 9-cis retinoic acid or DHA, is a highly debated and controversial issue [56]. Similarly, whereas all known HNF4 crystal structures are liganded, the biological significance of HNF4 ligands is not clear, since the latter are either non-exchangeable molecules found in the LBD structure or do not induce any transcriptional activity [57]. For SpNR1 and SpNR2, barely no information is available. Functional characterization of sponge receptors combined with phylogeny analysis and ancestral sequence reconstruction allowed Bridgham *et al.* to propose that NRs evolved from a ligand-activated ancestral receptor that existed near the base of the Metazoa, with fatty acids as possible ancestral ligands [9]. Taken together, these data indicate that the presence of the π-turn in RXR and HNF4 is not related to the ligand binding capability.

Interestingly, we observe that the absence of a π-turn is correlated with the loss of H7-H11 stabilizing bonds and as a consequence, is linked to a greater flexibility of the ligand binding site which is indeed partly composed of helices H7 and H11 [58]. Several examples of NR LBDs have been reported where the ligand binding pocket nicely adapts and molds to different types of ligands, by exhibiting remarkable changes in the structural elements composing the binding cavity. It is in particular the case for EcR [59], ER [60], VDR [61,62], PXR [63], and

many other NRs. For all of them, the adaptation of the pocket to the ligand occurs through substantial changes of the region encompassing helices H7 and H11, and the β-sheet. Focusing on helix H7, the structural adaptation of this helix can occur only when it is devoid of the structural constraints that would be imposed by the presence of a π-turn. In other words, molding and adaptability to various ligand molecules is correlated to the absence of a π-turn. Thus, from the evolutionary point of view, the disappearance of the π-turn in more recent nuclear receptors (from NR1 and NR4 subfamilies) facilitated the binding of a variety of molecules and promoted their diversification.

The presence of a π-turn in RXR and HNF4 suggests that its maintenance is linked to a different function, namely dimerization (Fig 7). Our analysis supports the key role of the π-turn of RXR in heterodimer formation through numerous interactions with the loop H8-H9 and the helices H9 and H10 of the partner NR. Experimental evidence provided here for the case of PPARα/RXRα LBD fully supports our hypothesis. The lack of π-turn in the partner receptor strengthens the resulting asymmetric heterodimer. We hypothesize that the presence of the π-turn in RXR is a necessary condition for this receptor to be the ubiquitous dimerization partner of many different NRs. This structural feature is namely linked to a stiffening of the LBD structure, especially the heterodimerization region, allowing RXR to dimerize in a similar fashion with different partner receptors.

## The π-turn represents an unusual exaptation

In protein science, it has always been thought that the π-turn is a structural feature that evolved in a way such as to accommodate novel functionalities. This is not the case here, since the π-turn is an ancestral motif that was instead lost during NR diversification. However, its presence or its absence is linked to critical biological functions. On the one hand, the presence of the π-turn in the most ancestral receptors is crucial for the stabilization of a homodimer interface in the context of small molecule binding in the LBD for sensor function. On the other hand, the loss of the π-turn in all subsequent NRs allowed their binding site to adapt to a different type of ligands and for a large group of them facilitated their heterodimerization with RXR in a stronger and asymmetric manner.

The origins of novelties still remain a central question in evolutionary biology. A fundamental question is how organisms constrained by natural selection can divert from existing schemes to set up novel structures or pathways. Among all possible strategies addressing this issue, pre-adaptations [64], which are also called exaptations, are one of the most important strategies. According to Gould and Vrba, exaptations are "*features that now enhance fitness but were not built by natural selection for their current role*" [65]. Numerous examples of exaptation have been proposed at the morphological and the molecular levels. Feathers used for bird flight originate indeed as thermoregulation devices in dinosaurs (they were also colored: reproductive and camouflage functions as well). At the molecular level, crystalline lenses first emerged as metabolic enzymes that were later on recruited in the eye for their light-refracting function [66]. More recently, cases of exaptation were identified in the case of retroviral envelope proteins that were recruited as placental proteins in early mammals [67] or in amphibians as a mechanism for functional reinforcement of a pheromone system [68]. Exaptations are furthermore frequently observed at the gene expression level, mainly through the recruitment of new gene regulatory elements allowing cooptation of gene function in novel organs, tissues or process. This is for example the case of the reinforcement of the courtship pheromone system in frogs via the co-option of the persuasin gene [68]. This has also been observed in several cases after gene duplication [69]. Exaptations are however less frequent for protein structures. One example are the bifunctional metabolic enzymes [70] or the sea urchin fibropellin protein, for

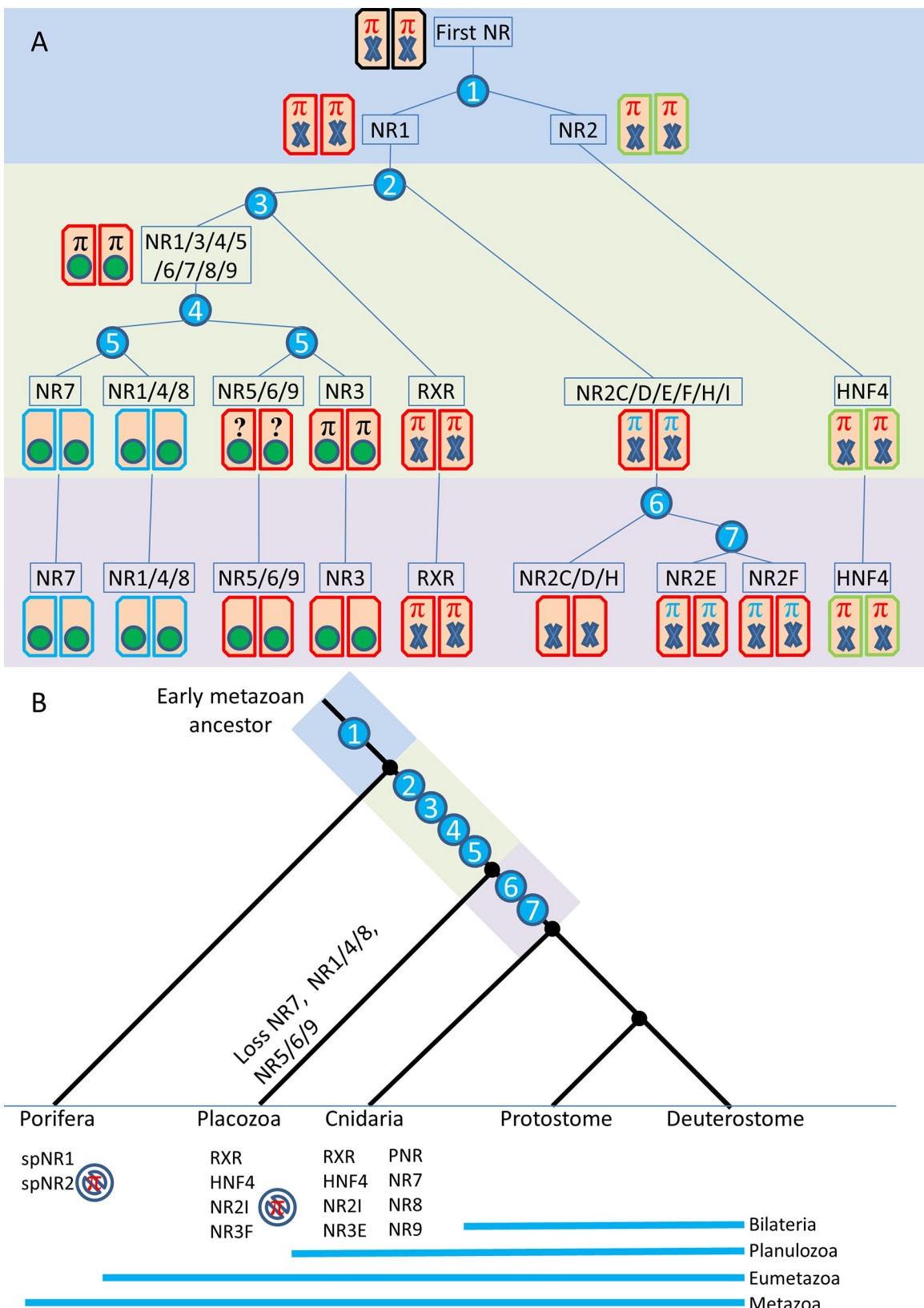

**Fig 7. Evolution of the π-turn motif and dimerization interface.** (A) Evolutionary history of the main receptor families in regard to π-turn and dimerization. The rectangles correspond to LBD monomers and their color indicates their class marker composition: red (e.g. RXR) for all Class I markers, green (e.g. HNF4) when the W109 and R321 are missing, and blue (e.g. NR7) for other Class I marker anomalies. The first NR is indicated in black as no information is available. The blue numbered circles are duplication events that are plotted on the tree on panel B. The ligand binding pocket is indicated as a green circle when it is liganded and a blue cross when there is no or weak non-specific interaction. The π-turn motif is indicated by a big PI symbol (π), red for motif and structure present, black for motif present but no information about structure, blue for motif present and structure absent and '?' symbol when no information is available. (B) Phylogenetic position of the evolutionary events described in (A) (blue numbered circles). For porifera, placozoans and cnidarians, the minimal sets of receptors present in their last common ancestor are indicated.

which a dimerization motif evolved from a biotin binding domain [71]. Here, we propose that the π-turn, an ancestral structural feature that was present in ancient receptors, in particular in RXR and HNF4, was important for homodimerization and later utilized by RXR as a key structural element for heterodimerization. However, its loss allowed the reinforcement of ligand adaptation in other NRs. Both contrasting aspects eventually led to a substantial expansion of the repertoire of NR regulatory abilities.

To summarize, both the presence (in RXR) and the absence (in NR partners) of the π-turn led to the emergence of new NR function, namely the heterodimerization of RXR with partner receptors, leading to greater target site selection and the emergence of high affinity receptors due to more flexible binding site that could diversify in terms of ligand binding possibilities. We propose that the π-turn in NRs represents a case of structural exaptation, namely a trait whose benefit for the system is unrelated to the reason of its origination, but which allowed an unprecedented increase of the NR regulatory repertoire.

## Materials and methods

### Structure refinement of COUP-TF LBD

A careful analysis of the crystal structure of COUP-TFII and its corresponding electron density map reveal that large portions of the electron density in the region of H7 could not be interpreted. The main problems are located between H5 and H7, with no visible electronic density for the β-sheet and H6 connecting H5 to a disordered N-terminal part of H7. For the latter a closer inspection to the electron density map suggests that the N-terminal part of H7 could adopt different conformations. Since this region was critical for our analysis of the RxxxE motif and the structural features associated to it, we decided to further improve the protein structure around this location by iterative building in Coot of residues in the non-interpreted electron density map followed by a crystallographic refinement using Phenix. This work resulted in better crystallographic quality factors R and Rfree and to a more confident interpretation of the electron density map (see S1 Table).

After crystallographic re-refinement, we observe that in the crystal packing helix H7 can adopt two helical structures, together with a lengthening of helix H7 at its N-terminal side as compared to the original helix of the PDB structure (S2 Fig). The two novel conformations correspond to a regular straight and a curved α-helix bent at the level of the putative π -turn. The C-terminal parts of the two helices overlap nicely, while their N-terminal ends are located over 6 Å apart. These conformations are in equilibrium in the crystal, alternating between nearest neighbour molecules to ensure optimal packing and are likely to be natural conformations. The dynamics of H7 resulting from the absence of a stabilizing H11 promotes the adaptability to packing constraints with a subsequent disorder of this subdomain. In fact, the lengthening of the original single helix H7 to the size of the re-refined one would lead to steric clashes between crystallographic dimers.

The second important observation is the absence of the π–turn. Although the electron density was not clear enough to confidently build the side chain of R293, some density can be seen that could correspond to the guanidinium group of the arginine in the straight conformation of the helix, indicating that in this conformation, the intra-helical salt bridge between the side chains of the arginine and the glutamic acid of the RxxxE motif could be maintained. However, this intra-helical salt bridge is rotated to a position such that no interaction between the H7 motif and H10-H11 can take place. The shift induced by the absence of the π -turn prevents E206 from binding R316, instead the connection is made with its neighboring residue Q207 (Q298 in hCOUP-TFII). The conserved serine residue of RXR H11 that stabilizes the π -turn in RXR-USP and HNF4 is replaced by a threonine residue, but without interacting with H7 residues. Furthermore, no interactions are seen between H7 and H5-H6. Of note, helix H7 after refinement does not exhibit a 3(10) helical turns as suggested in the original structure.

## Evolutionary analysis

Collected NR sequences were aligned using Clustal Omega (Sievers and Higgins, 2014) and alignments were checked manually and edited with Seaview (Gouy et al., 2010). Phylogenetic trees were built using PHYML (Guindon and Gascuel, 2003). Following model testing using AIC and BIC criteria as implemented in the SMS software (Lefort at al., 2017), we selected the LG model (Le and Gascuel, 2010) with a gamma law and estimation of the proportion of invariable sites. The reliability of nodes was assessed by likelihood-ratio test (Anisimova and Gascuel, 2006). Ancestral character reconstruction and stochastic mapping (Huelsenbeck et al., 2003) were performed under R version 3.2.2 (R Core Team, 2015) using the make.sim-map function as implemented in the phytools package version 0.5.0. (Revell, 2012). Character evolution was inferred using a model of symmetrical transition rates between the character states (SYM). 10 000 character histories were sampled to allow the incorporation of the uncertainty associated with the transition between different states. Inferred state frequencies for ancestral nodes were plotted using the describe.simmap function. Commands and sources files for ancestral mapping are provided in the Dryad repository under the following doi:10.5061/dryad.kkwh70s48.

## Cloning, expression and purification for biophysical studies

HsRXRα LBD, wild type (T223-T468) and mutants E352A and ΔE352, were cloned into the pET15b expression vector. HsPPARα LBD (I195-Y468) was cloned in a pET15b expression vector. Each individual vector was transformed into *Escherichia coli* BL21 (DE3), grown at 37˚C and induced for protein expression at an $OD_{600nm}$ of 0.6 with 1 mM IPTG at 25˚C for 3 hours. The corresponding cell pellet was resuspended in binding buffer (20 mM Tris pH = 8.0, 400 mM NaCl, 10% glycerol, 2 mM CHAPS, 5 mM imidazole) and lysed by sonication. The crude extract was centrifuged at 45'000 g for 1 hour at 4˚C. The lysate was loaded on a Ni affinity step on HisTrap FF crude column (GE Healthcare, Inc.) and the protein was eluted at a concentration of 150 mM imidazole. The LBD protein was then polished by size-exclusion chromatography in a SEC buffer (20 mM Tris pH = 8.0, 250 mM NaCl, 2 mM TCEP) by using a Superdex S75 16/60 column (GE Healthcare).

## Polyacrylamide native gel electrophoresis

The individual proteins were run on an 8% polyacrylamide gel (PAGE) at 2 W constant power after pre-running the gel for 40 min at 4˚C. The native gel system was based on a Tris/CAPS (pH = 9.4) buffer system that contained 60 mM Tris base and 40 mM CAPS (3-cyclohexil-amino-1-propane-sulfonic acid). Approximately 3–5 μg protein was loaded per lane along

with its DNA counterpart at defined molar ratios.The polyacrylamide gels were stained using Instant Blue Protein Stain (Expedeon Protein Solutions) for 15 min and rinsed in water.

## Analytical ultracentrifugation

Sedimentation velocity experiments were conducted using Beckman Coulter ProteomeLab XL-I analytical ultracentrifuge using the 8-hole Beckman An-50Ti rotor at 4˚C for samples in a buffer composed of 20 mM Tris pH = 8.0, 250 mM NaCl, 20 μM TCEP [72]. The molar protein concentration of the experiments corresponds to 1 μM. Sedimentation at 50000 rpm was monitored by absorbance at 220 nm with scans made at 5 min intervals. The solution density and viscosity for resuspension buffer were calculated using SEDNTERP software. Data were analyzed using a c(s) model in SEDFIT [73].

## Size-exclusion chromatography hyphenated to non-denaturing mass spectrometry (SEC-non denaturing MS)

For SEC-non-denaturing MS analysis, an ACQUITY UPLC H-class system (Waters, Manchester, UK) comprising a quaternary solvent manager, a sample manager cooled at 10˚C, a column oven maintained at room temperature and a TUV detector operating at 280 nm and 214 nm was coupled to the Synapt G2 HDMS mass spectrometer (Waters, Manchester, UK). 50 μg of each samples were loaded on the ACQUITY UPLC Protein BEH SEC column ($4.6 \times 150$ mm, 1.7 μm particle size, 200 Å pore size from Waters, Manchester, UK) using an isocratic elution of 150 mM ammonium acetate ($NH_4OAc$) at pH 7.4 and at a flow rate of 0.25 mL/min over 4.0 min. Then the flow rate was decreased to 0.10 mL/min over 5.9 min and finally increased to 0.25 mL/min over 1.9 min. The Synapt G2 HDMS was operated in positive mode with a capillary voltage of 3.0 kV while sample cone and pressure in the interface region were set to 40 V and 6 mbar, respectively Acquisitions were performed in 1,000–10,000 m/z range with a 1.5 s scan time. The mass spectrometer was calibrated using singly charged ions produced by a 2 g/L solution of cesium iodide (Acros organics, Thermo Fisher Scientific, Waltham, MA USA) in 2-propanol/water (50/50 v/v). Native MS data interpretations were performed using Mass Lynx V4.1 (Waters, Manchester, UK). For PPARα/RXRα experiments, the two nuclear receptors were expressed in E. coli and purified separately. Mixtures of 1:1 molar ratio of RXRα (wt or mutants) and PPARα LBD were performed in the purification buffer, incubated and injected on the SEC column coupled to the mass spectrometry instrument as described above. Deconvolution was performed using UniDEc [74]. Relative abundances of the species were calculated from native MS intensities of the deconvoluted data.

## Off-line native electrospray-mass spectrometry (ESI-MS)

Samples were first buffer exchanged in 150 mM ammonium acetate ($NH_4OAc$) at pH 7.4 using 0.5 mL Zeba$^{TM}$ Spin desalting Columns (Thermo Fisher Scientific, Waltham, MA USA). Then, concentrations were determined by UV-Vis using a Nanodrop 2000 Spectrophotometer (Thermo Fisher Scientific, Waltham, MA USA). Finally, analyses were performed on an electrospray time-of-flight mass spectrometer (LCT, Waters, Manchester, UK). Samples were diluted to a monomeric protein concentration of 10 μM and directly infused into the mass spectrometer via an automated chip-based nanoESI source (Triversa Nanomate, Advion, Ithaca, NY). Instrumental parameters were optimized for the detection of noncovalent complexes by raising the interface pressure to 6 mbar and the cone voltage to 60 V. Acquisitions were performed in 1,000–10,000 m/z range with a 4 s scan time in positive mode The mass spectrometer was also calibrated using singly charged ions produced by a 2 g/L solution of cesium iodide (Acros organics, Thermo Fisher Scientific, Waltham, MA USA) in 2-propanol/water (50/50 v/

v). Native MS data interpretations were performed using Mass Lynx V4.1 (Waters, Manchester, UK).

## Molecular modeling of sponge NR1

Homology modeling of the NR1 sequence from *Amphimedon queenslandica* (ACA04755.1) using multiple templates was performed using Modeler (Webb and Sali, 2016) in order to evaluate the possible three-dimensional fold of this sequence, in particular whether a π-turn could be formed. The multiple alignment was constructed using 15 PDB structures of RXR (1MV9, 1H9U), USP (1Z5X, 1G2N, 1HG4), HNF4 (1LV2, 1PZL), LRH (1PK5, 1YUC), RAR (1DKF, 1FCY, 1XAP), TR (1NAV), ERR (1S9P) and LXR (1UPV). Sequence identity between NR1 sequence and these templates ranged from 27% (ERR) to 42% (RXR). A total of one hundred models were generated and evaluated according to their DOPE scores. The model with the best score, as well as all other models obtained contain a π-turn in helix H7, with very similar orientations of R and E residues when compared to RXR (obtained in the structure PDB code 1DKF). Since sequence identity is very elevated between NR1 and RXR and HNF4 sequences, another homology model was built using the same multiple alignment with only sequences of receptors without a π-turn and consisted of LRH (1PK5, 1YUC), RAR (1DKF, 1FCY, 1XAP), TR (1NAV), ERR (1S9P) and LXR (1UPV). Similarly, a total of one hundred models were calculated and their quality evaluated according to their DOPE scores. In this case, the best model does not contain a π-turn in H7, however when comparing the scores of both best models from the two modeling strategies, the model with π-turn has the best score (-27875.7) with respect to the one without a π-turn (-27771).

Model assessment was also calculated using ProQ2 (Ray et al., 2012), an algorithm predicting local and global quality of protein models, based on properties from sequence (predicted secondary structure for example) and structure (atom-atom contacts, residue-residue contacts, secondary structure). This algorithm provides a score for each residue and was used to assess the quality of the homology modeling specifically for helix H7. We calculated the average score of helix H7 (ranging from 0 to 1, the latter being the best score) and obtained 0.71 and 0.65 with standard deviations of 0.06 and 0.04 for the models with and without π-turn respectively, supporting an enhancement in quality with the presence of the π-turn in the nuclear receptor structure.

Homology modeling using multiple templates was performed using Modeler (Webb and Sali, 2016), sequences with an e-value of 0 (best alignment) were extracted from non-redundant PDB sequences. The multiple alignment was constructed using 15 PDB structures of RXR (1MV9, 1H9U), USP (1Z5X, 1G2N, 1HG4), HNF4 (1LV2, 1PZL), LRH (1PK5, 1YUC), RAR (1DKF, 1FCY, 1XAP), TR (1NAV), ERR (1S9P) and LXR (1UPV). Sequence identity between NR1 sequence and these templates ranged from 27% (ERR) to 42% (RXR). A total of one hundred models were generated and evaluated according to their DOPE scores. Another homology modeling run was performed using structures without a π-turn in helix H7 and consisted of LRH (1PK5, 1YUC), RAR (1DKF, 1FCY, 1XAP), TR (1NAV), ERR (1S9P) and LXR (1UPV). Similarly, a total of one hundred models were calculated and their quality evaluated according to their DOPE scores.

## Molecular Dynamics simulations

Simulations were performed with the LBD of RXR using the crystal structure 1MVC. Substitution of E352 residue by an alanine in the wild-type structure was performed using the mutagenesis wizard in the PyMol program (The PyMOL Molecular Graphics System VS, LLC.). Generation of the ΔE352 was done using Modeler, providing the structure of RXR as template

for the ΔE352 sequence. A model without a π-turn was thus generated, and we insured that all other side chains aside from this region remained similar to the wild-type structure. Hydrogen atom placement was performed using the HBUILD facility [75] in the CHARMM program [76]. All three structures were solvated in cubic boxes of approximately 121 per side, with a salt concentration Na+/Cl- corresponding to the physiological concentration of 150mM. Before solvating the system, two minimizations of 100 steps of Steepest Descent method and 1000 steps of Adapted Basis Newton-Raphson method were performed in order to eliminate steric clashes.

Molecular dynamics simulations were performed using the CHARMM36 force field [77] within the NAMD program [78], following two steps. First, minimization and heating of water molecules around the fixed protein was realized with 1000 steps of Conjugate Gradient (CG) energy minimization, heating up to 600K over 23ps, 250 steps of CG energy minimization, and heating to 300K over 25ps. Second, positional restraints on the protein were removed and all the system was energy minimized with 2000 steps of CG and heating to 300K over 15ps, followed by 85ps of equilibration. The production run was then performed for the duration of 50ns. Periodic boundary conditions were used and the particle mesh Ewald algorithm [79] was applied to take into account long-range electrostatic interactions. All bonds between heavy atoms and hydrogens were constrained using the SHAKE algorithm [80] and an integration time step of 2fs was used for all simulations. This protocol has been carefully benchmarked across different nuclear receptor proteins, such as RAR, ER, GR [81–85]. The first 10ns of the production run were excluded from all analysis to ensure proper equilibration. Time evolution of $C_\alpha$-RMSD of the three systems wt RXRα LBD, ΔE352 RXRα LBD, and E352A RXRα LBD across the three simulations are represented in S6D Fig to illustrate the stability of the structures over the course of the analyzed timeframe.

Binding free energies were estimated on the average structures, calculated over the time frame between 10 to 50ns. We used the University of Houston Brownian Dynamics (UHBD) [86], to solve the linearized Poisson-Boltzmann equation and compute the electrostatic binding free energy of binding of the two molecules. A dielectric constant of 80 was used for the solvent, 1 for the protein and a pH of 7. Van der Waals radii and charges of atoms are obtained from the force field CHARMM36. A nonbonded cutoff of 12.5 Å were used with a shift truncation function for electrostatics. Although MM/PBSA does not take into account conformational entropy, our protocol [87] has been validated on a variety of systems to assess protein/protein complexes in terms of relative free energies and proven to be in agreement with experimental data [84,85].

## Supporting information

**S1 Fig. Reference sequence alignment snippet used in this article.** Alignment generated with Clustal Omega software and manually corrected.
(TIF)

**S2 Fig. Helix H7 of the ligand-binding domain of COUP-TFII displays a double conformation.** (A) Ribbon diagram showing the double conformation of helix H7. Conformation A is colored in green, and conformation B in blue in all images. All images are presented in cross-eye stereo. (B) 2mFo-DFc electron density map contoured at 0.5 sigma (0.16 e-/Å$^3$) around helix H7. The density for conformation B is shown in blue, and the supplemental density for conformation A is shown in green. (C) 2mFo-DFc electron density map contoured at 0.5 sigma (0.16 e-/Å$^3$) around conformation B only of helix H7. (D) 2mFo-DFc electron density map contoured at 0.5 sigma (0.16 e-/Å$^3$) around conformation A only of helix H7.
(TIF)

**S3 Fig. Maximum-likelihood phylogenetic tree of nuclear receptors.** Classical and newly defined NR families are indicated with grey boxes. Species sequences are colored according to the five main metazoan groups they belong to. Branch support values are assessed by approximate likelihood ratio test (aLRT) and are show only for nodes that are considered fully robust (above 0.97). Occurrence of the RxxxE motif is also indicated. Our improved sampling also confirms that there are two cnidarian-specific NR groups. The first one is NR8, that branches at the basis of the crow group containing the bilaterian-specific NR1/NR4 group. The NR1/4/8 group is sister to the eumetazoan NR7 group, which has undergone lineage specific losses in vertebrates and arthropods, and is therefore represented here by sequences from the leech *Helobdella robusta* and *Branchiostoma lanceolatum*. Although the internal branching between NR1, N4, NR7 and NR8 are not supported by strong branch support, they are consistent with the last global phylogenetic analysis of the NR family [9]. In this study, the NR7 group was informally called "INRa" and the NR7 group was called "INRb". Also, there is an additional cnidarian-specific group branching at the basis of the bilaterian NR5 and NR6 groups, which was unformally called "cSF-1". Here it is named NR9, since it is at the basis of NR5 and NR6, and two additional sequences from the anthozoan cnidarians *Pocillopora damicornis* and *Acropora millepora* are provided. Concerning the NR3 family, cnidarian sequences were identified [88] and here, the Hydra sequence was used as a representative of this group. Interestingly, among basal groups, the loss of the RxxxE motif is always associated with other deviations from the ancestral state. In the NR2A group, the RxxxE motif has been lost in a group comprising the Ctenophoran sequences, that completely lost their DBD ((Reitzel et al., 2011); in blue, as well as in all of the sponges sequences, in which the DBD diverges through a 2 to 5 residue insert. Inside the sponge SpNR1 group, it was lost once in the paralog P3 group which exhibits the longest branch length, further indicating of a high substitution rate. In the NR2E group, the RxxxE motif is fully lost in the longest branch leading to Drosophila FAX-1, whereas among the other branches, there are already indications of higher structural variation. There is a KxxxE motif in Drosophila DSF and Nematostella 183874, a QxxxE motif in humain TLX, and an RxxxxxxE motif in Drosophila TLL. On the other hand, the RxxxE motif is conserved in the PNR group, for bilaterian and cnidarians, and also in the two NR2E1 sequences from cnidarians with the shortest branches (S2 Fig). In the NR2F group, only one cnidarian sequence, the one of *Hydractinia echinata*, is fully lost the RxxxE motif, while it is present as the KxxxE variant in two more cnidarian sequences. The 21656 sequences from the placozoan *Trichoplax adhaerens*, which branches separately from the eumetazoan NR2C/D/H, NR2E and NR2F group, illustrate the fact that early diverging organisms may also deviate from the ancestral pattern. Lastly, regarding the crown group containing NR5/6/9, NR3 and NR1/4/7/8, the Trichoplax NR3 sequence was the only one to keep the RxxxE motif.
(TIF)

**S4 Fig. Oligomeric status of wild-type (wt) RXRα LBD and mutants, where the conserved Glu residue of the π-turn motif is mutated to Ala (E352A) or deleted (ΔE352).** (A) Size-exclusion chromatograms from analytical SEC chromatography on a S200 10/300 Superdex column for wt RXRα LBD (olive line), E352A RXRα LBD (cyan line) and ΔE352 RXRα LBD (grey line). Peaks are named void for the peak roughly corresponding to the exclusion volume of the column, 'peak1 wt' and 'peak2 wt' for the two peaks of wt RXRα LBD and 'peak mut' for the peak of the mutant constructs. The peaks are further marked by symbols in colors that are then consistently used for the panel B; red for peak1 wt, blue for peak2 wt, cyan for E352A RXRα and grey for ΔE352 RXRα. The polyacrylamide SDS gel depicted in the insert indicates the high purity of the samples. (B) Differential sedimentation coefficient distribution c(S) as a

function of the sedimentation coefficient S (in Svedberg) obtained from AUC experiments at 1μM, showing the c(S) distribution for peak 1 wt (red curve) and peak 2 wt' (blue curve) of the wt RXRα LBD SEC peaks, as well as for peak mut of E352A RXRα LBD SEC peak (cyan curve) and ΔE352 RXRα LBD SEC peak (grey curve).
(TIF)

**S5 Fig. Oligomeric status of wild-type (wt) RXRα LBD and mutants, where the conserved Glu residue of the π-turn motif is mutated to Ala (E352A) or deleted (ΔE352).** (A-D) Native mass-spectrometric analysis of RXR LBD wt and mutants performed at a voltage 60 V of the SEC peaks. The inserts on the right show an enlarged view (by a factor of 6) of the m/z region that encompasses the dimer m/z peaks centered around the 16+ species. Striking is the complete absence of tetramers for the RXR LBD mutant species. (E) Native polyacrylamide gel electrophoresis of the different SEC peaks of wt RXRα LBD (peak 1 and peak2) and of the mutant E352A RXRα LBD and ΔE352 RXRα LBD. T and M specifies the tetrameric and the monomeric species, respectively. (F) Cartoon representation of tetrameric RXRα as observed in the crystal structure of RXRα LBD (PDBcode 1G1U). The two homodimers of the tetramer (A1/B1 and A2/B2) are shown in yellow and cyan, respectively. The Arg and Glu residues of the π-turn are shown as sticks, colored in magenta for carbon, blue for nitrogen and red for oxygen. The triple Phe residues of H11 are depicted in as sticks colored in grey for carbon. Important helices of the tetramer interface are indicated.
(TIF)

**S6 Fig. Electrostatic interactions for the RXR wt and mutant homodimers.** The electrostatics interactions were analyzed during the MD simulations of (A) wt, (B) E352A and (C) ΔE352 RXRa LBD homodimer complex. The percentage of interaction presence is represented with lines connecting the corresponding residues, in red (between 75 and 100% of simulation time), purple (between 50 and 75% of simulation time) and pink (between 25 and 50% of simulation time). (D) Time evolution of the Ca-RMSD of the wt RXRα LBD (black line), E352A RXRα LBD (blue line) and ΔE352 RXRα LBD (red line), represented in black, red and blue respectively, over the course of the 50ns molecular dynamics simulations.
(TIF)

**S7 Fig. Heterodimeric dimerization interface for EcR/USP-RXR of different species.** (A-B) Detailed views of heterodimeric interactions between EcR and USP-RXR LBD structures of different insect species, the more recent species *Heliothis virescens* (Hv), and the more basal species *Tribolium castaneum* (Tc) and *Bemisia tabaci* (Bt). The structure of HvEcR/USP-RXR is the one bound to 20-hydroxyecdysone (PDBcode 2R40), that of TcEcR/USP-RXR is bound to ponasterone A (PDBcode 2NXX), that of BtEcR/USP-RXR is bound to ponasterone A (PDBcode 1Z5X). In (A), the view shows the interactions of USP-RXR H7 with the EcR partner loop H8-H9 which are rather conserved among species. In (B), the view shows the large differences in interaction distances between EcR H7 helix and the partner USP-RXR at the level of its loop H8-H9. Intermolecular interactions between EcR H7 and USP-RXR H8-H9 loop can only be formed for HvEcR/USP-RXR and not for TcEcR/USP-RXR and BtEcR/USP-RXR, due to too large distances between these structural elements.
(TIF)

**S8 Fig. Heterodimerization capacity of wild-type (wt) RXRα LBD and mutant E352A and ΔE352 RXR LBD with PPARalpha.** Size-exclusion chromatography (SEC)-coupled native mass spectrometric (MS) analysis for complex mixture of PPARα LBD with either (A) wt RXRα LBD, (B) E352A RXRα LBD or (C) ΔE352 RXRα LBD. The isolated RXRα LBD (wt or mutants) is depicted in yellow, isolated PPARα LBD in blue and heterodimeric PPARα/RXRα

LBDs in green. The raw mass spectra are depicted together with the size-exclusion chromato-grams of wt RXRα LBD and of mutants RXRα LBD. Integration of the mass spectra is per-formed over the whole peak of the chromatogram.
(TIF)

**S1 Table. Data collection and refinement statistics.** For the original structure, statistics for the highest resolution shell are given in parentheses. For the original structure reported statis-tics are shown in red, and recalculated statistics in blue.
(XLSX)

**S2 Table. Molecular masses measured in the size-exclusion chromatography (SEC)-cou-pled native mass spectrometric analysis.** Molecular masses are given for the two SEC-MS peaks seen for wt RXRα LBD and for the single peak observed for the mutants E352A RXRα LBD and ΔE352 RXRα LBD. The region of the SEC-MS peaks that were integrated for mass determination are shown in the insert of Fig 6B by a line over the peaks.
(XLSX)

**S3 Table. Molecular masses measured in the native mass spectrometric analysis.** Molecular masses are given for the different SEC peaks collected during the analytical SEC analysis shown in Fig 6A.
(XLSX)

**S4 Table. Total binding free energy from Molecular Dynamics (MD) simulations.** Shown is the total binding free energy between each RXR subunit of the homodimer made of wt RXRα LBD, E352A RXRα LBD and ΔE352 RXRα LBD, respectively, calculated from 50 ns MD simu-lations.
(XLSX)

## Acknowledgments

The authors would like to thank Bariza Blanquier, manager of Genetic Analysis facility for her expert assistance (AniRA platform, SFR BioSciences Gerland—Lyon Sud UMS3444/US8). We would like to acknowledge the High-Performance Computing Center of the Université de Strasbourg for providing access to computing resources. We thank I. Hazemann for excellent technical assistance, Anthony Ehkirch for SEC-MS analysis, C. Birck for AUC experiments, K. Essabri for mutagenesis of HsRXR LBD, B. Klaholz and A. Dejaegere for discussion and Fran-çois Bonneton for review and comments. We thank Bruno Frédérich (Université de Liège) for help in using phytools.

## Author Contributions

**Conceptualization:** Brice Beinsteiner, Gabriel V. Markov, Vincent Laudet, Dino Moras, Isa-belle M. L. Billas.

**Data curation:** Brice Beinsteiner, Gabriel V. Markov, Yassmine Chebaro, Alastair G. McEwen, Isabelle M. L. Billas.

**Formal analysis:** Brice Beinsteiner, Gabriel V. Markov, Alastair G. McEwen, Sarah Cianférani, Vincent Laudet, Dino Moras, Isabelle M. L. Billas.

**Funding acquisition:** Sarah Cianférani, Vincent Laudet, Dino Moras, Isabelle M. L. Billas.

**Investigation:** Brice Beinsteiner, Gabriel V. Markov, Stéphane Erb, Yassmine Chebaro, Alas-tair G. McEwen, Isabelle M. L. Billas.

**Methodology:** Brice Beinsteiner, Gabriel V. Markov, Yassmine Chebaro, Sarah Cianférani, Isabelle M. L. Billas.

**Project administration:** Vincent Laudet, Dino Moras, Isabelle M. L. Billas.

**Resources:** Brice Beinsteiner, Gabriel V. Markov, Sarah Cianférani.

**Software:** Brice Beinsteiner, Gabriel V. Markov, Yassmine Chebaro.

**Supervision:** Sarah Cianférani, Vincent Laudet, Dino Moras, Isabelle M. L. Billas.

**Validation:** Brice Beinsteiner, Gabriel V. Markov, Yassmine Chebaro, Sarah Cianférani, Vincent Laudet, Dino Moras, Isabelle M. L. Billas.

**Visualization:** Brice Beinsteiner, Gabriel V. Markov, Alastair G. McEwen, Sarah Cianférani, Vincent Laudet, Dino Moras, Isabelle M. L. Billas.

**Writing – original draft:** Brice Beinsteiner, Gabriel V. Markov, Vincent Laudet, Dino Moras, Isabelle M. L. Billas.

**Writing – review & editing:** Brice Beinsteiner, Gabriel V. Markov, Sarah Cianférani, Vincent Laudet, Dino Moras, Isabelle M. L. Billas.

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
