## [Decision Letter · Decision Letter 0]

15 Mar 2021

Dear Dr Billas,

We are pleased to inform you that your manuscript entitled "A Structural Signature Motif Enlightens the Origin and Diversification of Nuclear Receptors" has been editorially accepted for publication in PLOS Genetics. Congratulations!

Yours sincerely,

Jianzhi Zhang

Associate Editor

PLOS Genetics

Kirsten Bomblies

Section Editor: Evolution

PLOS Genetics

Comments from the reviewers (if applicable):

Reviewer's Responses to Questions

**Comments to the Authors:**

Reviewer #1: The authors adequately addressed my comments from the initial review at Review Commons.

Reviewer #2: The authors have addressed all my concerns satisfactorily.

Reviewer #3: Beinsteiner et al. have revised of their manuscript to meet my earlier comments. Moreover, they have adequately responded to the critiques of the other reviewers.

This revision is a major improvement, proving clarity on the role of the �-turn in the evolution of heterodimers in nuclear receptors.

**Have all data underlying the figures and results presented in the manuscript been provided?**

Reviewer #1: Yes

Reviewer #2: Yes

Reviewer #3: Yes

PLOS authors have the option to publish the peer review history of their article (what does this mean?). If published, this will include your full peer review and any attached files.

Reviewer #1: No

Reviewer #2: No

Reviewer #3: No

**Data Deposition**

http://datadryad.org/submit?journalID=pgenetics&manu=PGENETICS-D-20-01883

**Press Queries**

---

## [Editor Report · Acceptance letter]

13 Apr 2021

PGENETICS-D-20-01883 

A Structural Signature Motif Enlightens the Origin and Diversification of Nuclear Receptors 

Dear Dr Billas, 

We are pleased to inform you that your manuscript entitled "A Structural Signature Motif Enlightens the Origin and Diversification of Nuclear Receptors" has been formally accepted for publication in PLOS Genetics! Your manuscript is now with our production department and you will be notified of the publication date in due course.

With kind regards,

Katalin Szabo

PLOS Genetics

On behalf of:
